# Molecular Basis of Na, K–ATPase Regulation of Diseases: Hormone and FXYD2 Interactions

**DOI:** 10.3390/ijms252413398

**Published:** 2024-12-13

**Authors:** Bárbara Martins Cordeiro, Carlos Frederico Leite Fontes, José Roberto Meyer-Fernandes

**Affiliations:** 1Instituto de Bioquímica Médica Leopoldo de Meis, Centro de Ciências da Saúde, Universidade Federal do Rio de Janeiro, Rio de Janeiro 21941-590, RJ, Brazil; barbara.cordeiro@bioqmed.ufrj.br; 2Instituto Nacional de Ciência e Tecnologia em Biologia Estrutural e Bioimagem, Rio de Janeiro 21941-590, RJ, Brazil

**Keywords:** Na, K–ATPase, FXYD2, ion transport, hormonal regulation, signal transduction, receptors, protein kinases, diseases, hypertension, renal hypomagnesemia, cancer

## Abstract

The Na, K–ATPase generates an asymmetric ion gradient that supports multiple cellular functions, including the control of cellular volume, neuronal excitability, secondary ionic transport, and the movement of molecules like amino acids and glucose. The intracellular and extracellular levels of Na^+^ and K^+^ ions are the classical local regulators of the enzyme’s activity. Additionally, the regulation of Na, K–ATPase is a complex process that occurs at multiple levels, encompassing its total cellular content, subcellular distribution, and intrinsic activity. In this context, the enzyme serves as a regulatory target for hormones, either through direct actions or via signaling cascades triggered by hormone receptors. Notably, FXYDs small transmembrane proteins regulators of Na, K–ATPase serve as intermediaries linking hormonal signaling to enzymatic regulation at various levels. Specifically, members of the FXYD family, particularly FXYD1 and FXYD2, are that undergo phosphorylation by kinases activated through hormone receptor signaling, which subsequently influences their modulation of Na, K–ATPase activity. This review describes the effects of FXYD2, cardiotonic steroid signaling, and hormones such as angiotensin II, dopamine, insulin, and catecholamines on the regulation of Na, K–ATPase. Furthermore, this review highlights the implications of Na, K–ATPase in diseases such as hypertension, renal hypomagnesemia, and cancer.

## 1. Introduction

The Na, K–ATPase, also known as the sodium pump, is a transmembrane enzyme that, under physiological conditions, mediates the hydrolysis of one intracellular ATP molecule into ADP and orthophosphate, exchanging three Na^+^ ions in the extracellular space for two K^+^ ions in the intracellular space. This sodium pump is essential for all mammalian cells; at rest, it consumes 20–30% of the energy derived from ATP. An electrochemical gradient, at a ratio of 3:2:1 (Na:K:ATP), is necessary for maintaining the membrane potential, cell volume, and active and secondary transport of other solutes, such as glucose, amino acids, and other ions. Additionally, it facilitates transcellular transport in the intestine, adrenal glands, and kidneys [1,2].

Since its discovery, Na, K–ATPase has been the subject of numerous studies using its specific inhibitor, ouabain (OUA) [3]. These studies demonstrated that Na, K–ATPase not only functions as an ion transport enzyme but also functions in cellular signal transduction, leading to cell differentiation and proliferation, enzyme turnover, and the control of gene expression [4,5,6]. However, the mechanisms of hormonal action and the pathways by which the enzyme acts as a signal transducer to relay messages through protein–protein interactions from the plasma membrane to the nucleus remain poorly understood.

The aims of this review are to correlate the myriad of functional changes in the Na, K–ATPase structure upon hormonal regulation and the actions of protein kinases and FXYD proteins to identify the potential implications of these enzyme modifications in health and pathophysiology. It is primarily important to us to describe in good detail the molecular mechanisms underlying the events which lead to the regulation of the Na-pump to shift from a homeostatic physiological status becoming a player in the origin and development of several diseases.

## 2. Na, K–ATPase: Structure and Function

The Na, K–ATPase belongs to the P-type ATPase class, within which five main families have evolved to transport a wide variety of substrates across different biological membranes. The Na, K–ATPase and H, K–ATPase form the 2C subfamily and are hetero-oligomers with obligatory binary αβ complexes that couple the export of H^+^ or Na^+^ to the import of K^+^ in animals. The catalytic α subunits possess the characteristic central structure of P-type ATPases, with 10 transmembrane helices (TMs) (~110 kDa), which are responsible for ion transport [7]. Additionally, these proteins include a smaller (~31.5 kDa) and highly glycosylated β subunit that plays a vital role in ensuring the correct folding and anchoring of the α subunit on the cell surface, acting as a chaperone. This subunit also modulates the ion affinity of the enzyme and plays a fundamental role in cell adhesion [8]. In certain tissues, the α-β heterodimer is associated with one of the seven members of the FXYD protein family, which modulate the kinetic properties of the enzyme [9]. In the kidneys, FXYD2, a small polypeptide also known as the γ subunit (~7 kDa), is purified with classical outer medulla Na, K–ATPase preparations [10,11,12,13]. The assembly diagram in Figure 1 highlights each subunit of the Na, K–ATPase.

During the catalytic cycle, the enzyme undergoes phosphorylation and dephosphorylation, with the transported cations becoming temporarily occluded within the intermembrane portion. The currently accepted Albers–Post model postulates that the enzyme in its nonphosphorylated form (E1) has a high binding affinity for one ATP molecule and three Na^+^ ions on the intracellular surface in the presence of Mg^2+^ [14]. The binding of three Na^+^ ions to the E1ATP intermediate triggers the phosphorylation of an aspartate residue, leading to the trapping of Na^+^ ions within the newly formed phosphoenzyme (E1-P). The release of ADP into the cytoplasmic is followed by a spontaneous conformational change (E2-P), in which the binding sites are exposed to the exterior. The affinity for Na^+^ ions is reduced to such an extent that they are released into the extracellular medium. In contrast, the binding sites gain a high affinity for K^+^ ions; afterward, two K^+^ ions bind, the extracellular portion closes, and the aspartate residue is dephosphorylated. During the dephosphorylation process, the E1-P form of the enzyme still holds an ADP molecule and two Na^+^ ions; the transition from E1-P to E2-P culminates with the release of ADP into the cytoplasm and the exposure of external K^+^ binding sites, which is essential for enzymatic activity [15]. Shortly thereafter, the binding of K^+^ to the external sites accelerates the dephosphorylation of the E2-P intermediate, culminating in the release of Pi from the discharge site. The E2(K^+^) intermediate is very stable, making the release of K^+^ into the intracellular space the rate-limiting step of the reaction cycle. The binding of a new ATP molecule induces a conformational change to E1, decreasing the affinity of the pump for K^+^, opening the intracellular portion, and consequently leading to the release of the ions (Figure 2). At this point, the pump is ready to initiate a new catalytic cycle.

The existence of Na^+^-containing intermediates in the Na, K–ATPase under noninhibited conditions demonstrates the presence of strongly bound Na^+^ in an enzyme capable of undergoing a catalytic cycle. Additionally, the spontaneous occlusion of Na^+^ in the dephosphorylated E1 form of the enzyme indicates positive interactions between the binding sites, challenging the conventional understanding that intracellular Na^+^ must bind to the transport sites and be occluded after ATP phosphorylation to be transported to the extracellular medium. These findings reveal that Na^+^-occluded species decrease transiently during the course of ATP hydrolysis, which is related to the formation of E2-P and the disappearance of preexisting Na^+^-occluded E1 forms. These results, therefore, provide significant evidence of the dynamics of Na^+^ occlusion and its relationship with the reaction cycle of Na, K–ATPase [16], establishing the current understanding of the transport mechanism of this enzyme.

The sodium and potassium electrochemical gradients maintained by the Na, K–ATPase are essential for animal physiology. In most cells, both excitable and nonexcitable, the resting membrane potential is largely determined by the diffusion of sodium, potassium, and chloride ions, and this balance is maintained by the ATP-driven activity of the Na, K–ATPase [17]. When functioning electrogenically with a 3Na^+^:2K^+^ ratio, the enzyme directly influences the electrical potential across the membrane, although this effect is minimal in cells with high potassium permeability [18,19,20]. Under physiological conditions, the enzyme cannot pump ions against their electrochemical gradient beyond certain negative membrane potentials (approximately −50 mV for a 4:3 ratio and −70 mV for a 4:2 ratio). Therefore, at a typical resting membrane potential of −85 to −75 mV, the pump would not function with those stoichiometries. However, at a 3:2 ratio, ATP hydrolysis provides sufficient energy for Na^+^ and K^+^ transport at positive membrane potentials up to −100 mV. This finding suggests a thermodynamic limitation of the stoichiometry of the Na, K–ATPase, which is likely influenced by the electronegative intracellular environment and the energy derived from ATP [21].

## 3. Regulatory Mechanisms

The regulation of Na, K–ATPase is a complex process that occurs at multiple levels, including the total cellular content, subcellular distribution, and intrinsic activity. Local regulatory mechanisms primarily involve the concentrations of Na^+^ and K^+^ inside and outside the cell, as well as factors such as hypoxia, purines, oxidative stress, pH, nitric oxide, and ATP, which all influence Na, K–ATPase activity [22].

Systemically, hormones play a central role by regulating enzyme expression and activity on the cell surface through phosphorylation by protein kinases [23,24,25]. The translocation of Na, K–ATPase from the intracellular compartments to the plasma membrane is controlled by the phosphorylation of the α-subunit, a posttranslational modification [26,27,28]. Furthermore, this α-subunit contains several serine, threonine, and tyrosine residues that can be phosphorylated by different kinases, thereby influencing Na, K–ATPase activity [29].

The phosphorylation of the Na, K–ATPase α-subunit by kinases is an important regulatory mechanism. In this context, the regulation of activity and ion transport occurs in both short- and long-term responses to hormones and kinases. Hormones that control the expression and activity of Na, K–ATPase include angiotensin II, catecholamines such as dopamine, epinephrine, and norepinephrine, as well as thyroid hormones and insulin. These hormones, as illustrated in Figure 3, are also implicated in complex responses both in vivo and in vitro [30,31,32].

However, the cellular transducers of these actions are kinases, indicating a strong correlation between specific Na, K–ATPase conformations and PKC autophosphorylation [33]. The PKA phosphorylates Ser 943 in the α1 isoform at a 1:1 ratio relative to ATPase. On the other hand, PKC can phosphorylate both Ser and Thr residues (in the N-terminal region) at a ratio of 2 moles of phosphate/mol of ATPase. In both systems, in vitro phosphorylation reduced activity by up to 50% [34,35]. In experiments involving the controlled expression of different α isoforms in *Xenopus oocytes*, isoform-specific phosphorylation was reported, showing that only α1 was phosphorylated at a regulatory level by PKC [36]. The modulation of Na, K–ATPase by PKC appears to be conserved across different species, for example, even in a protozoans such as *Leishmania amazonensis* [37] through a pathway involving PI–PLC/PKC [38]. Moreover, it has been demonstrated that receptor tyrosine kinases (RTKs), such as those in the insulin and epidermal growth factor (EGF) cascades, phosphorylate Na, K–ATPase, activating hydrolysis [39].

FXYDs (FXYD1–11) are small transmembrane regulators of Na, K–ATPase that are defined by the distinctive motif Phe-Xaa-Tyr-Asp (FXYD) [40]. While the sequences of FXYDs vary beyond the signature motif, their function seems to be preserved throughout the vertebrate lineage. In fact, all mammalian FXYDs are involved in regulating Na, K–ATPase [41]. Furthermore, FXYDs establish a connection between hormones and Na, K–ATPase at different levels. First, hormones acutely influence the interaction between Na, K–ATPase and FXYD; hormones regulate the expression of FXYDs and can act on different signaling pathways that may cause changes in enzymatic activity [42,43,44].

## 4. Regulation of Na, K–ATPase by FXYD2

FXYD2, also called the γ subunit, was originally thought to be the third component of the Na, K–ATPase complex. The γ subunit exists in two splice variants, γa (7.184 kDa) and γb (7.338 kDa), which are found in the kidneys and pancreas [45,46,47]. Research by Kuster et al. (2000) revealed that these variants differ in seven N-terminal residues, γa, TELSANH, are substituted by Ac-MDRWYL in γb, but the rest of the sequence is the same [48]. The localization of FXYD2a and FXYD2b variants was described in the literature along with nephron [49,50,51,52,53,54]. Both forms are predominantly expressed in the thick ascending limb and similarly reduce the apparent sodium affinity of Na, K–ATPase [39,46]. The FXYD2a and FXYD2b variants appear to undergo different posttranslational modifications that affect enzyme kinetics differently, although the physiological purpose of having two isoforms remains unclear [9,45]. The kinetic effects of the two variants are very similar, the works suggest that both decreased the apparent sodium affinity [55]. Consequently, it causes a reduction in Na, K–ATPase activity [47,56]. The expression of FXYD2 is restricted to the kidney and pancreas, and posttranslational modifications and effects on Na, K–ATPase are poorly known. The localization of FXYD2a and FXYD2b variants has been described in the literature along with nephron [49,50,51,52,53,54]. Some studies demonstrate that mutation in FXYD2b affects the regulation of the cell surface and cell growth [57]. The physiological and pathological role of FXYD2a and FXYD2b in the kidney remains unknown. However, for the pancreas, genomic-based research has proposed FXYD2a as a novel beta cell-specific biomarker. Histological examinations of pancreatic sections from individuals with type-1 diabetes, as well as sections from streptozotocin-treated monkeys, have revealed a correlation between the loss of FXYD2a expression and a reduction in insulin-positive cells. Consequently, FXYD2a has been proposed as a potential biomarker, with its application offering a valuable tool for monitoring cellular mass under various conditions, such as type 1 diabetes [58].

FXYD2 is not expressed in erythrocytes; however, the treatment of Na, K–ATPase from pig erythrocytes (ghosts) with FXYD2 extracted from the renal Na+ pump results in increases in the ATP hydrolysis activity of approximately 54% and 66% in the presence of low and high ATP concentrations, respectively. Moreover, FXYD2 also causes an increase in Ca+ATPase activity in pig erythrocytes (ghosts), resulting in increased activity in the presence of calmodulin [59]. Thus, in addition to various members of the FXYD family interacting with and regulating Na, K–ATPase, the FXYD2 from the sodium pump is capable of modulating the activity of other P-type ATPases.

Several studies have highlight the regulation of Na, K–ATPase by different FXYDs, such as FXYD10, CHIF, FXYD7, FXYD2, and FXYD1 [60,61]. Therefore, it is possible to infer that the anchoring site of FXYD2 on the α subunit is likely highly conserved and matches with numerous FXYD isoforms, allowing cross-interactions between them. Mutagenesis studies have shown that the residues Phe956, Glu960, Leu964, and Phe967 in the M9 loop of the α subunit of rat Na, K–ATPase are important for the formation of a stable complex with FXYD [62].

A denaturation study suggested that the γ subunit might interact with the M8–M10 region [63]. Recently, a role for M9 of the α subunit was inferred from the effects of M9 mutants on the stability of α/β-γ, α/β-CHIF, or α/β-FXYD7 complexes and their functional consequences in *Xenopus* oocytes [64]. In this context, crystallography data obtained in the presence of vanadate and OUA indicate that the M9 loop may be the domain of the α subunit that interacts with FXYDs [62,65]. The multiple alignment of the N-terminal portion of the amino acid residues between the M8 and M10 transmembrane segments of the α subunit of rat and crustacean Na, K–ATPase revealed the interaction of the Phe956, Leu964, and Phe967 residues in the M9 loop with FXYD2 [13].

In 1999, in vitro experimental assays, using Na, K–ATPase purified from the pig renal outer medulla, demonstrated that the Oubain Binding Promoting Factor (OBPF) is a proteolipid that was later shown to be the FXYD2. FXYD2 promotes the binding of OUA to the sodium pump [66,67].

When the γ subunit is depleted from the Na+ pump in the renal medulla, its activity is inhibited by approximately 35% [59], but the activity increases after a preincubation with the previously extracted γ subunit. Some effects of γ subunit modulation on Na, K–ATPase have been described, particularly with respect to changes to Na^+^ and K^+^ affinities [46,68]. Past studies have highlighted the role of the ATP regulatory site in the effects of the γ subunit on the hydrolytic reaction [47]. When Na, K–ATPase was saturated by the addition of a molar excess of FXYD2, an increase in the maximum velocity was observed at both the catalytic and regulatory substrate sites. However, no significant differences in the affinity for ATP of the enzyme were noted [59].

In addition to the significant reduction in activity and restoration upon the re-addition of previously extracted FXYD2, Cortes et al. (2006) observed greater enzyme activation when the γ subunit is phosphorylated by PKA and PKC, with no additional effect when the subunit is simultaneously phosphorylated by both PKA and PKC. The phosphorylation sites of the FXYD2 peptide by PKA and PKC were investigated [60]. Western blot analysis using antibodies against phosphoserine, phosphothreonine, and FXYD2 indicated that FXYD2 is endogenously phosphorylated by PKA. When it is doubly phosphorylated by PKA and PKC or solely by PKA at exogenous levels, it strongly reacts with the antiphosphoserine antibody. When FXYD2 is phosphorylated only by exogenous PKC, a strong reaction with the antiphosphothreonine antibody is observed, while no phosphorylation is detected at endogenous levels. Moreover, this study showed that the FXYD2 peptide can be phosphorylated only by protein kinases in association with the α and β subunits [60].

Furthermore, the α subunit is also phosphorylated by PKA, as observed in SDS gels, but only in the presence of Triton X-100. In the absence of this detergent, a considerable decrease in phosphorylation levels is observed [35]. The phosphorylation of the γ subunit by PKA without Triton X-100 is reduced but still detectable [67]. Thus, the activation of the Na+ pump through the effects of FXYD2 and the modulation of phosphorylation by endogenous and exogenous kinases highlights the role of FXYD2 in signal transduction.

## 5. Cardiotonic Steroids

Cardiotonic steroids (CTSs), also known as cardiac glycosides, were first documented by ancient Egyptians more than 3000 years ago [69]. CTSs are widely recognized toexert therapeutic effects on treating congestive heart failure, which led to the coining of the term cardiotonic [70]. CTSs are classified into two main categories: cardenolides, such as OUA and digoxin, and bufadienolides, including marinobufagenin, telocinobufagin, and bufalin [71]. The positive inotropic effects of cardiotonic steroids, which contribute to their therapeutic benefits in patients, are primarily linked to their high-affinity inhibitory interaction with the extracellular surface of the sodium pump [72]. However, CTSs were initially recognized for their crucial roles in regulating renal sodium transport and controlling arterial pressure. The Na, K–ATPase also plays a role in various intracellular signaling pathways and serves as a receptor for CTSs, which can trigger alterations in intracellular signaling upon binding to the enzyme [73].

The mechanism of action of cardiac glycosides is based on the partial inhibition of Na, K–ATPase coupled with a reduction in Na^+^/Ca^2+^ exchanger activity, consequently increasing intracellular Ca^2+^ levels [74]. At therapeutic doses, OUA increases the strength of heart contractions. In the 1990s, endogenous OUA was detected in the bovine hypothalamus and the bloodstream of mammals at nanomolar concentrations. OUA has long been used medicinally to treat congestive heart failure and arrhythmia. Exogenous administration of OUA at micromolar concentrations results in the partial, specific inhibition of the Na, K–ATPase pump function, which is the basis for its cardiotonic effect but also accounts for its toxicity at these concentrations [75,76].

OUA is a unique cardenolide due to its high polarity, largely because it contains five hydroxyl groups in its steroid structure, which makes it more water-soluble than digitalis-based cardenolides. As a result, OUA is more hydrophilic and does not easily cross cell membranes unless it is bound to Na, K–ATPase, where it enters cells through a slower process of endocytosis. Once inside, it is recycled out of the cell via the lysosomal pathway [77]. In contrast, other CTSs, such as digoxin, which have fewer hydroxyl groups (only two in the case of digoxin), can penetrate cells more readily. While the sugar components of CTSs are not essential for Na, K–ATPase binding or ion pump inhibition, they significantly slow the dissociation of OUA and digoxin once they are bound to the enzyme, stabilizing their interactions [78].

The inhibition of N, K–ATPase by OUA occurs after the phosphorylation of the sodium pump with the binding of Na^+^, Mg^2+^, and ATP, which alters the conformation to the phosphoenzyme form (E1-P). In this state, the enzyme binds OUA at a single site on the α subunit, resulting in a 50% reduction in ATPase activity [79]. When OUA was added to the K^+^-sensitive phosphoenzyme in the E2-P conformation, the inhibition was more pronounced, indicating that OUA preferentially binds to the E2-P conformation, at a single class of sites and produces a stable complex that is not dephosphorylated [80].

One of the effects of OUA is to stimulate the E-P of Na, K–ATPase via Pi in the presence of Mg^2+^, increasing the affinity of the enzyme for Pi. OUA binding depends on Mg^2+^ [81]. Additionally, molecular modeling studies revealed that the rhamnose residue of OUA is involved in five intermolecular hydrogen bonds with the α1 subunit of Na, K–ATPase, which is crucial for the activation of Na, K–ATPase signaling after the binding of OUA. The main contributor to the binding energy of OUA is its steroid core, which forms a series of hydrogen bonds and hydrophobic interactions with Na, K–ATPase, stabilizing the ligand–receptor complex. The rhamnose residue directly interacts with four Na, K–ATPase residues, all of which are notably charged (Glu116, Glu312, Arg880, and Asp884), forming five hydrogen bonds with Glu312, Arg880, and Asp884 [82].

Moreover, cardiotonic steroids exert a broad range of pharmacological effects, such as cardiotonic, antiarrhythmic, antidiabetic, immunomodulatory, antibacterial, antifungal, antiprotozoal, antiviral, antineoplastic, sleep-inducing, analgesic, contraceptive, and endocrine activities, along with effects on behavior and wound healing [83]. Consequently, focusing on the Na, K–ATPase pump has become a priority for various research groups aiming to elucidate the pharmacological impacts of CTS, including their cardiotonic, anticancer, anti-inflammatory, and antiviral properties [84,85,86,87].

## 6. Vanadate

Vanadate is a phosphate transition state analog that stabilizes an E2 conformation of the Na, K–ATPase, mimicking the E2-P or E2^•^ Pi conformation, it acts as a blocker at the phosphate discharge site [88,89]. The Thr214→Ala mutations of the TGES conserved motif of the phosphorylation site displayed a conspicuous 151-fold reduction in the apparent vanadate affinity during catalytic turnover in the ATPase assay [90]. Vanadate is one of the most famous and studied P-type ATPase inhibitors. However, even four decades since this discovery, several POMs (Polyoxometalates) are still considered very promising drugs for the treatment of several diseases, including cancer. Any disease in which Na, K–ATPase, SERCA, or H, K–ATPase can be ascribed as the main target would have a POM as a drug choice for potential treatment [90]. The decavanadate is considered a potent mixed-mode inhibitor of P-type ATPases that alters the kinetics of Na, K–ATPase from rat synaptosomes (IC50 = 470 nM). It can inhibit the enzyme decreasing the maximum velocity and the apparent affinity for ATP, thus increasing K_m_ for substrate [91].

The decavanadate is considered a potent mixed-mode inhibitor of P-type ATPases that alters the kinetics of Na, K–ATPase from rat synaptosomes (IC_50_ = 470 nM). It can inhibit the enzyme decreasing the maximum velocity and the apparent affinity for ATP, thus increasing K_m_ for substrate [92].

## 7. Oxidative Stress

The oxidative stress is related to an increase in the generation of reactive oxygen species (ROS); ROS cause several effects on the Na, K–ATPase, and this has been extensively described in the literature. Oxidative stress causes many modifications in proteins, like s-glutathionylation, s-nitrosylation and carbonylation, among others. The elevated levels of ROS induce the oxidative modification of the Na, K–ATPase α and β subunits alongside FXYD proteins [93,94,95,96,97]. These oxidative modifications caused by superoxide, nitrite peroxide, S-nitrosoglutathione (GSNO), and other free radicals donors can irreversible inhibit Na, K–ATPase through phospholipid peroxidation at vicinity of the catalytic site [98]. The lipid peroxidation impaired Na, K–ATPase through modifications of the α1 subunit in the kidney exacerbating the dissipation of monovalent cation gradients [99,100]. Coronary artery disease (CAD) patients had higher levels membrane lipid peroxidation that were negatively correlated with decreased Na, K–ATPase activity. On the other hand, they were positively correlated with the severity of CAD [101].

Hyperglycemia induces an increase in endothelial superoxide that inhibits the stimulatory effect of nitric oxide (NO) on vascular Na, K–ATPase activity [102]. Nitrite peroxide is the ROS product of the reaction of NO and superoxide. In addition, it is a potent inhibitor of Na, K–ATPase activity, and it can induce amino acid modifications to the pump [103,104].

Notably, the S-glutathionylation of the β1 subunit, along with the associated inhibition of enzyme activity, can be dynamically reversed by FXYD proteins [94]. Research has shown that ROS are pivotal in the Na, K–ATPase signaling pathway. ROS induces modifications in the α1 subunit of Na, K–ATPase, leading to the activation of Src’s tyrosine kinase activity. This activation initiates a signaling cascade that amplifies ROS production and modulates other pathways, including the Ras/Raf/MEK/ERK1/2 cascade, mitochondrial ROS generation, PLC/PIP2/inositol triphosphate, PLC/PIP2/diacylglycerol/protein kinase C, and PI3K/AP2/clathrin-mediated endocytosis, along with their downstream effects [105]. Consequently, the Na, K–ATPase signaling cascade appears to function as a feed-forward mechanism that amplifies ROS, with circulating cardiotonic steroids modulating its intensity. Evidence from cellular and animal disease models indicates that this amplification mechanism is activated under oxidative stress conditions, such as obesity/metabolic syndrome, chronic kidney disease, cardiovascular disease, cancer, and other related disorders commonly linked to oxidative imbalance. Therefore, it presents a promising therapeutic target for clinical intervention [106,107].

The relationship between oxidative stress and the enzymatic interaction with FXYD proteins varies depending on the isoform expressed and the cell type involved in different pathologies. FXYD1 has been identified as playing a crucial protective role in vascular health, particularly by shielding the Na, K–ATPase from oxidative inhibition. Silencing FXYD1 in human umbilical vein endothelial cells led to a 50% reduction in nitric oxide (NO) production and a twofold increase in superoxide levels. Additionally, knockout mice exhibited heightened oxidative and nitrosative stress, along with impaired vascular function, especially under diabetic conditions. FXYD1 protects endothelial NO synthase from redox dysregulation, offering protection against both hypertension and diabetic vascular oxidative stress [108]. In this regard, Cai et al. (2023), using molecular biology tools, genetic engineering, and metabolic analyses to investigate the role of the Na, K–ATPase (NKA)-Src axis in mitochondrial function in human-induced pluripotent stem cell-derived cardiomyocytes (hiPSC-CMs), demonstrated that Na, K–ATPase α1/Src regulation plays a role in the tonic stimulation of mitochondrial metabolism and ROS production. This has potential implications for addressing mitochondrial dysfunction in cardiometabolic diseases [109].

Most FXYD proteins protect the β1 subunit against glutathionylation, an oxidative modification that destabilizes the heterodimer and inhibits Na, K–ATPase activity. The point mutation in the cysteine residue of FXYD3 increases sensitivity to oxidative stress induced by the chemotherapeutic doxorubicin and γ-irradiation [110,111].

## 8. Angiotensin II

Angiotensin II (Ang II), an octapeptide, has potent vasopressor effects, but its action is swiftly terminated by its degradation into angiotensin III via the angiotensinases present in red blood cells and the vascular beds of most tissues.

In addition to its systemic effects on blood pressure, fluid, and electrolyte homeostasis, Ang II contributes to the regulation of the volume status and vascular tone through both the tissue-specific and intracellular renin–angiotensin–aldosterone system (RAAS), which is present in most organs. The tissue RAAS is involved primarily in local cardiovascular regulation and inflammatory processes, including vascular permeability, apoptosis, cell growth, migration, and differentiation, whereas the intracellular RAAS participates in signaling pathways that further influence these cellular responses. Together, these systems ensure the precise control of cardiovascular and renal functions.

Ang II is hydrolyzed into Ang (1–7) by the enzyme ACE2 [112]. Ang II is crucial in the RAAS, as it regulates sodium transport in the epithelial cells of the kidney; acts as an endocrine, paracrine, and autocrine factor; and is reported as an anti-natriuretic hormone [113,114]. The regulatory mechanisms of Na^+^ and water excretion occur through intracellular signaling pathways capable of inhibiting or activating Na, K–ATPase [115,116]. Na, K–ATPase generates the electrochemical gradient necessary for reabsorption in the proximal tubule through stimulation by Ang II [117,118]. Initially, the stimulatory effect of Ang II on Na, K–ATPase activity was only thought to be indirect [119]. Later, Ang II was shown to acutely stimulate Na, K–ATPase activity by rapidly increasing the amount of Na, K–ATPase in the plasma membrane via a trafficking mechanism mediated by the PKC-dependent phosphorylation of the Na, K–ATPase at S11 and S18 [120,121].

Ang II receptors are characterized into two main subtypes based on their sensitivity to antagonists and coupling to G proteins: angiotensin type 1 receptor (AT1R) and angiotensin type 2 receptor (AT2R). Ang II binds to AT1R, triggering several classic signaling pathways. The activation of phospholipases A2 and D by Ang II results in the production of prostaglandin E2. It inhibits adenylate cyclase, reducing cyclic AMP levels and contributing to the vasoconstrictive response. Additionally, it activates phospholipase C (PLC), leading to the release of calcium from intracellular stores and subsequent vasoconstriction, while also promoting the activation of PKC. Through an unknown mechanism, Ang II activates Janus kinases (JAKs) and signal transducers and activators of transcription (STATs), which phosphorylate important cellular regulators to modulate the expression of early growth response genes. Finally, Ang II opens calcium channels, allowing calcium entry that results in the secretion of hormones (e.g., aldosterone) and vasoconstriction [122].

The signaling induced by Ang II via the AT2R receptor is mediated by the activation of protein tyrosine and serine/threonine phosphatases. These enzymes dephosphorylate other regulatory cellular proteins, promoting vasodilation, apoptosis, and differentiation and inhibiting proliferation. The activation of the AT2 receptor also stimulates the production of bradykinin, which generates nitric oxide. This process increases cyclic guanosine monophosphate (cGMP) levels, contributing to the vasodilatory response.

Ang II alters the conformation of Na, K–ATPase, modifying its interaction with cardiotonic steroids [123]. Additionally, Ang II induces the phosphorylation of the Ser11 and Ser18 residues of the Na, K–ATPase α subunit through a signaling pathway involving PKC-β and the interaction of Na, K–ATPase with adaptor protein 1 (AP1). This process results in the recruitment of Na, K–ATPase molecules to the plasma membrane (Figure 4), increasing enzymatic activity, which, in turn, leads to antinatriuretic effects and increased Na^+^ and water reabsorption [124].

Moreover, in transgenic mice with a variant of G protein-coupled receptor kinase 4 (GRK4), an increase in AT2R phosphorylation and a reduction in sodium excretion capacity were observed compared with mice without the variant. This increased phosphorylation of AT2R was associated with the reduced inhibition of Na, K–ATPase. The interaction between GRK4 and AT2R is stronger in hypertensive rats than in control rats, and when GRK4 expression is reduced, AT2R phosphorylation decreases, restoring diuresis and natriuresis in these animals [121]. Therefore, the mechanisms of action of Ang II and GRK4 appear to involve the regulation of Na, K–ATPase (Figure 4), influencing renal function and sodium homeostasis, particularly under hypertensive conditions.

Studies using proximal tubules from rats have revealed the biphasic regulation of Na, K–ATPase by angiotensin II: lower concentrations of Ang II (10^−13^–10^−10^ M) stimulate activity, whereas higher concentrations (10^−7^–10^−5^ M) inhibit the sodium pump [125]. The inhibitory effect is related to the cAMP accumulation observed in preparations with relatively high concentrations of Ang II, suggesting a relationship with the adenylate cyclase–cAMP–PKA pathway, as the phosphorylation of the α subunit by PKA inhibits Na, K–ATPase activity (Figure 4).

In contrast, preparations with low concentrations of Ang II do not exhibit altered basal cAMP levels but show inhibition dependent on cAMP accumulation stimulated by forskolin (1 µM) [126]. Compared with AT2R, AT1R is generally the predominant receptor; AT2R expression levels are lower than those of AT1R in most adult tissues. AT2R is expressed in tissues such as the kidney, heart, vasculature, adipose tissue, and brain [127]. Thus, the effects of Ang II are likely linked to AT1R activation. Despite the suggested effects of Ang II on Na, K–ATPase activity, the modulatory mechanisms are still not well understood.

## 9. Dopamine

Non-neuronal dopamine (DA) acts in the kidneys to regulate salt metabolism. In the kidney, the precursor of dopamine, L-dihydroxyphenylalanine (L-DOPA), is absorbed by proximal tubular cells from the ultrafiltrate [128], which contains a high concentration of the enzyme L-aromatic amino acid decarboxylase (AADC) [129]. AADC converts L-DOPA into dopamine, which acts as an autocrine and paracrine natriuretic hormone along the nephron [130].

Dopamine receptors are generally divided into two main subtypes that are characterized by their ability to either stimulate (D1) or suppress (D2) adenylyl cyclase, and both subtypes are present in the kidney. The physiological actions of dopamine are mediated by five distinct but closely related G protein-coupled receptors (GPCRs) [131]. In the central nervous system, D1 receptor functions depend on a dopamine- and cAMP-regulated phosphoprotein known as DARPP-32, which has a molecular weight of 32,000 Daltons. DARPP-32 cycles between its active, phosphorylated form and its inactive, dephosphorylated state; it is influenced by protein kinases, such as PKA, and phosphatases, such as calcineurin [115,132]. Active DARPP-32 specifically inhibits protein phosphatase 1 (PP1), channeling signals from both D1 and D2 receptors to this target. In the kidneys, phosphorylated DARPP-32 reduces Na, K–ATPase activity through this mechanism. D1/D2 heteromers coupled to G proteins are associated with calcium signaling through phospholipase C (PLC) transduction. The activation of PLC results in the generation of inositol trisphosphate (IP3), which, by activating its IP3 receptor (IP3R), triggers the release of calcium from the endoplasmic reticulum. This process also leads to the formation of diacylglycerol (DAG), which activates protein kinase C (PKC).

The activation of PKA and PKC by DA inhibits Na, K–ATPase activity by phosphorylating the Ser18 residue of the α subunit [133]. Experiments using opossum kidney cells have shown that DA activates phosphatidylinositol-3 kinase through a PKC-ζ-dependent pathway. This activation cascade leads to the subsequent internalization of sodium pump molecules, inhibiting their enzymatic activity and sodium reabsorption, and, consequently, producing natriuretic effects [134]. Moreover, studies using LLCPK1 cells have suggested an alternative pathway for renal sodium regulation, in which L-DOPA significantly inhibits Na, K–ATPase activity in an AADC-dependent manner [135].

## 10. Balance Between Ang II and DA: A Dynamic Equilibrium

Some studies suggest that the sodium pump is bidirectionally regulated by the balance between Ang II and DA. Sodium excretion is regulated by both DA through the activation of D1 receptors and Ang II through the activation of AT1R receptors. Exposure to a D1R agonist results in the rapid partial internalization of AT1R receptors and complete inhibition of signaling generated by these receptors. Similarly, exposure to Ang II leads to the rapid partial internalization of D1R receptors and the inhibition of downstream signaling. The results of this study indicate that AT1R and D1R function as multiprotein complexes with the opposing mechanisms of action in the regulation of the sodium pump in renal cells, providing a highly versatile and sensitive system for the short-term regulation of sodium excretion [136,137]. Experiments using renal tissue from Sprague–Dawley rats demonstrated that Ang II inhibits DA uptake via AT1R [138]. Therefore, Ang II and DA modulate the other’s effects at the renal level. Experiments using transfected human embryonic kidney 293T (HEK-293T) cells have shown that AT1R and D2R can form heterodimers that exhibit specific signaling and cross-antagonism [139].

In human proximal tubular renal cells, sodium reabsorption is decreased by DA via the activation of D1-like receptors (D1R/D5R) and by Ang II via the activation of the AT2R receptor. The activation of D1-like dopaminergic receptors stimulates increased AT2R expression in the plasma membrane through a specific signaling pathway coupled to cAMP and dependent on protein phosphatase 2A [140]. Moreover, the stimulation of D1R/D5R receptors reduces the effect of Ang II on extracellular phosphate levels, a signal necessary for the regulation of kinase proteins. This effect was partially reversed by AT2R antagonists [141]. D1R and AT2R act cooperatively, increasing cAMP and cGMP production, activating phosphatase 2A, internalizing the sodium pump, and inhibiting sodium transporters. Additionally, the stimulation of the D5R receptor reduces the function and expression of AT1R in rodents and humans. The activation of D1R receptors increases D1R and AT2R expression while reducing AT1R expression [141,142].

D1R and AT2R receptors are both natriuretic and are stimulated by DA and angiotensin III, respectively, inhibiting renal reabsorption and resulting in natriuresis [140]. Ang II is converted into Ang III by aminopeptidase A, which activates AT2R receptors, decreasing sodium transport [143]. Whether the local conversion of Ang II to Ang III can be regulated by the local production of DA under conditions of increased sodium levels is still unknown. The activation of dopaminergic and/or Ang II receptors, individually or simultaneously, is necessary for maintaining renal sodium homeostasis and controlling blood pressure.

## 11. Epinephrine

Epinephrine, a primary stress hormone, has been shown to influence water transport across epithelial layers in certain tissues, including the kidneys [144], human eyes [145,146,147], and lungs [148]. The transport of water across the epithelial layers in the colon is regulated by the Na^+^ gradient established by Na, K–ATPase [145].

The current literature provides limited insights, with some reports describing a stimulatory effect of epinephrine on the Na, K–ATPase pump in rat jejunal crypt cells [149] and skeletal muscle tissue [150], although its precise mechanism remains poorly understood. In Caco-2 cells, epinephrine treatment dose- and time-dependently decreased Na, K–ATPase activity. This inhibitory action, mediated by α2 adrenergic receptors, triggers a cascade involving Src, p38MAP, ERK, and COX-2, and ultimately leads to the release of prostaglandin E2 (PGE2), which acts specifically via EP1 receptors [145].

This pathway also involves PKC, as the inhibition of PKC with calphostin C prevented the epinephrine- and PGE2-mediated inhibition of Na, K–ATPase, and the activation of PKC by PMA replicates this inhibitory effect on the Na, K–ATPase, confirming that PKC is downstream of PGE2 [151].

Furthermore, PKC activation stimulates the transcription factor NF-κB, which increases the expression of inducible nitric oxide synthase (iNOS) [152] increasing nitric oxide (NO) production [153]. NO is known to modulate Na, K–ATPase via two distinct pathways: cGMP-dependent and cGMP-independent mechanisms. The cGMP-independent pathway involves NO interacting with superoxide to generate free radicals, causing lipid oxidation and disrupting protein functions through the nitrosylation or nitration of key amino acids [154,155]. Through the cGMP-dependent pathway, NO activates soluble guanylyl cyclase, increasing cGMP levels and consequently activating PKG. This PKG activation inhibits Na, K–ATPase through the phosphorylation of intermediate signaling molecules [151]. This study also revealed that PGE2 decreases Na, K–ATPase protein expression in both cell homogenate and membrane fractions and consequently decreases total Na, K–ATPase activity without phosphorylation at tyrosine 10 of the α-subunit, indicating that PGE2 affects only ATPase expression levels [151].

Furthermore, epinephrine has a tropic effect on cardiac tissue, likely stimulating cell growth through the modulation of the signaling function of Na, K–ATPase via β1-adrenoreceptors. This effect appears to be related to the nonpumping function of Na, K–ATPase, which, when interacting with catecholamines (such as epinephrine and norepinephrine), operates in a receptor-coupled and regulated manner [156]. Additionally, previous studies have indicated that OUA also regulates the growth of cardiac explants in a dose-dependent manner, suggesting roles for Na, K–ATPase in growth signaling and the response of cardiac tissue to hormone stimulation [157].

## 12. Norepinephrine

Norepinephrine (NE), a catecholamine that primarily acts on α-adrenergic receptors, is frequently utilized in the management of patients experiencing septic shock when fluid resuscitation is insufficient to re-establish arterial blood pressure, particularly in patients with hypotension [158,159]. Various studies have characterized the relationship between Na, K–ATPase activity, and the effects of NE. The effect of NE is characterized by tissue-specificity [30,160,161,162,163,164]. The effects of NE generated by norepinephrine depend on the different subcellular fractions of Na, K–ATPase. The microsomal fraction, particularly the enzymes from the synaptosome, exhibits significant modulation, whereas the effect is absent on enzymes in the mitochondria, vesicle, and myelin fractions [165].

The activity of Na, K–ATPase in different brain regions revealed distinct sensitivities to NE. This difference is supported by the uneven distribution of synapses across various brain regions. At 0.01–1 mM, NE inhibited Na, K–ATPase in synaptosomes [165].

Experiments using Wistar rats indicated that NE stimulated Na, K–ATPase activity in different regions of the brain, such as the cerebral cortex, cerebellum, diencephalon, striatum, brainstem, and mesencephalon. The activities ranged from 8 to 16 µmol Pi/mg of protein/h. The percentage of activation over the basal level of the enzyme varied in each region, with the highest values observed in the cerebral cortex (128%) and the lowest in the striatum (36%). The other regions presented activation (~50%) [166].

Studies involving the activity of erythrocyte membrane Na, K–ATPase have demonstrated that 71% of patients with congestive heart failure exhibit lower Na, K–ATPase activity than control patients do, which is associated with increased plasma norepinephrine levels. The plasma norepinephrine concentration is higher in patients with nonsustained ventricular tachycardia than in those without ventricular tachycardia [162].

The impact of heart failure on pulmonary edema was investigated in Sprague–Dawley rats. This work presented evidence that the activity of Na, K–ATPase in peripheral lung tissues is increased in rats with chronic left atrial hypertension. Significantly elevated levels of norepinephrine were observed in the plasma of rats with arteriovenous fistulas, which may have contributed to the positive regulation of the active sodium transport system, thereby keeping the alveolar spaces free of fluids [167]. These effects are attributed to the activation of the cyclic AMP-mediated pathway and protein kinase A, which recruits Na, K–ATPase subunits within 15 min ofthe stimulation of the adrenergic receptor, as well as to the MAPK/ERK-dependent pathway [168,169]. Additionally, NE enhances active sodium transport and promotes the reabsorption of alveolar fluid in an isolated perfused rat lung model. This effect aligns with the observed increases in Na, K–ATPase activity and protein levels in isolated alveolar epithelial type II cells [161].

Several renal effectors are produced by the kidney following changes in Na^+^ intake, including effectors that elicit either natriuretic or anti-natriuretic responses [128]. One of these effectors, NE, stimulates Na^+^ reabsorption when dietary Na^+^ levels decrease, which in turn activates Na, K–ATPase activity in the renal proximal tubule (RPT), leading to increased Na^+^ reabsorption [160,164].

## 13. Thyroxine

Thyroxine (T4) and its more physiologically active form T3 are hormones that regulate metabolism and are produced by the thyroid gland [170]. The administration of thyroxine has protective effects on induced nephrotoxicity. It is associated with the increased activity of the Na, K–ATPase in the basolateral membranes of renal tubules [171,172,173,174,175].

Even at low doses, T3 is associated with increased passive K^+^ efflux from liver slices. This property is correlated with pump stimulation [176]. On the other hand, the sodium handling is severely compromised in the brains of mice lacking a thyroid gland [177]. A downregulation of sodium current density throughout the nervous system can explain the reduction in neuronal excitability seen in severe hypothyroidism.

Skeletal muscle is the tissue with the highest amount of Na, K–ATPase [178]. T3 stimulates Na, K–ATPase activity in the skeletal muscle via two regulatory mechanisms. First, indirectly, it allows the escape of Na^+^ and K^+^ ions, which impacts the homeostasis of the pump. Second, it increases the expression levels of isoform alpha2-beta1 of Na, K–ATPase [25]. However, an increase in the alpha2-beta2 isoform was observed in rats. The expected changes in pump content when subjects condition changes from hypothiroidsm to hyperthyroidism pump levels can be enhanced up to six times. This huge increase can be reverted if the animal is treated and its gland function is normalized to an euthyroid status. In summary, tyrosine, through its derivative hormones T4 and T3, plays an essential role in regulating Na, K–ATPase activity across various tissues, promoting protective and stimulatory effects on the enzyme. The influence of T3 on Na, K–ATPase activity has broad implications for energy metabolism and renal function, suggesting that thyroid hormones are vital for supporting enzymatic activity and the tissue adaptation to metabolic variations.

## 14. Insulin

Research has indicated that insulin stimulates solute and water reabsorption in the proximal convoluted tubule (PCT) of mammals by increasing active sodium transport, possibly by regulating Na, K–ATPase activity [179,180,181]. Studies have also shown that this activation by insulin is independent of PKC and is blocked by the tyrosine kinase inhibitor genistein but is replicated by the tyrosine phosphatase inhibitor orthovanadate, suggesting the involvement of tyrosine phosphorylation [182].

The phosphorylationof Tyr-10 in the cytoplasmic N-terminal tail of the α1 subunit of Na, K–ATPase is essential for its activation by insulin in renal proximal tubule cells. Without phosphorylation at Tyr-10, the Na, K–ATPase response to insulin is reduced, highlighting the importance of this residue in enzymatic activity [183].

Insulin increases alveolar fluid reabsorption and Na, K–ATPase activity by promoting the translocation of the enzyme to the plasma membrane in alveolar epithelial cells, a process responsible for clearing pulmonary edema from alveolar spaces and, consequently, allowing optimal gas exchange [184]. In this process, insulin-induced AKT activation is crucial and sufficient to direct Na, K–ATPase to the membrane. The phosphorylation of Asp160 by Akt is also necessary, whereas the inactivation of the Asp160 Rab GTPase-activating protein domain enables partial Na, K–ATPase translocation even in the absence of insulin. Rab10 has been identified as a downstream target of Asp160 in insulin-mediated Na, K–ATPase translocation. Together, these findings suggest an essential role for AKT in the intracellular translocation of Na, K–ATPase and alveolar fluid reabsorption [28].

Compared with lean individuals, the activity of Na, K–ATPase in the adipose tissue of obese patients is lower [185]. In diabetic rats, a decrease in Na, K–ATPase activity is also observed in the liver, while antidiabetic compounds can restore this activity in affected hepatic tissues [186]. Obesity, which is associated with hyperglycemia and hyperinsulinemia, can suppress or inactivate Na, K–ATPase [187]. Compared with control mice, the activity of Na, K–ATPase in the liver is reduced by 63% in mice characterized by hyperglycemia and hyperinsulinemia [188]. Similarly, mice fed a high-fat diet (HFD) exhibit reduced Na, K–ATPase activity and the decreased expression of the Na, K–ATPase α1 protein in the liver [189].

In rat hepatocytes and some skeletal muscle cells, the Na, K–ATPase can be rapidly activated by insulin, which stimulates the Na^+^/H^+^ exchanger. This fast increase in internal Na^+^ is the basis of the Na, K–ATPase stimulation because of the well-known sensitivity of the Na^+^ pump to changes in internal [Na^+^], which are rate-limiting in many tissues. In addition, this insulin effect is promptly blocked by amiloride [190]. Na, K–ATPase activity may also be regulated in the longterm where insulin promotes α1 and α2 expression or even reduces the degradation of the α1 subunit [191].

The decrease in Na, K–ATPase activity and alterations in its isoforms may contribute to cardiac dysfunction, myocardial dilation, and heart failure [192,193,194]. Additionally, diabetic cardiomyopathy is closely related to the impairment of Na, K–ATPase activity induced by advanced glycation end-products (AGEs) [195].

Other regulatory effects of insulin on Na, K–ATPase are observed in tissues such as the skeletal muscle, the brain, vascular smooth muscle, and tumors, as described in the review by Wen and Wan (2021) [196].

## 15. Diseases

### 15.1. Hypertension

The relationships between certain hormones that regulate Na, K–ATPase activity and hypertension are discussed in this section. First, endogenous OUA, a hormone produced in small amounts in the body, is associated with blood pressure regulation [80,197,198,199].

Endogenous OUA contributes to cardiovascular [200] and renal damage [201] and functions as a vasoconstrictor [202]. Furthermore, the plasma concentrations of endogenous OUA are positively linked to blood pressure [200,203]. Lowering the body’s total sodium content leads to an increase in circulating levels of endogenous OUA [204].

One of the molecular mechanisms associated with hypertension is the decreased activity of OUA-sensitive Na, K–ATPase in vascular smooth muscle. In dogs, OUA-sensitive Na, K–ATPase results in normal 86Rb uptake in normotensive animals, whereasits uptake is reduced by 42% in arteries and by 49% in the veins of hypertensive animals, suggesting that the dysfunction of OUA-sensitive Na, K–ATPase contributes to the elevated vascular resistance characteristic of hypertension [205]. Staehr et al. (2023) demonstrated the association between hyperactivation and the ouabain-stimulated overexpression of Na, K–ATPase in hypertensive models. Additionally, they highlighted protective mechanisms against hypertension in the alpha two isoform through changes in ouabain sensitivity [206]. Endogenous ouabain (EO) increases blood pressure (BP) binding to Na, K–ATPase, which subsequently activates cSrc/cSrc/EGFR/Raf/ERK1/2/p38MAPK. EO antagonist rostafuroxin prevents cSrc activation, improving endothelial activity while reducing oxidative stress and BP. EO acts as a crucial factor in end-organ damage in volume-dependent hypertension [207]. Renal denervation causes reduction in EO levels, Na, K–ATPase activity, and the alpha 1 subunit of kidney in spontaneously hypertensive rats [208]. For additional insights into the association of EO and its implications for hypertension, readers can refer to additional review articles on this topic [197,206].

Under normal conditions, renal dopamine helps prevent increases in blood pressure by inhibiting Na, K–ATPase. However, oxidative stress impairs the function of the D1Rs, resulting in mild hypertension. When oxidative stress is combined with high salt intake, severe hypertension occurs, as dopamine fails to promote sodium excretion due to D1R dysfunction [209]. Studies have indicated that impaired dopamine receptor function may play a role in the development of hypertension in both animal models and humans with essential hypertension [210,211].

The expression of the sodium–hydrogen exchanger regulatory factor 1 (NHERF-1) in renal proximal tubule cells is associated with the dysfunctional regulation of Na, K–ATPase by dopamine in hypertension models (SHR rats and aged F344 rats). Under normal conditions, NHERF-1 interacts with the α subunit of Na, K–ATPase and D1R, allowing dopamine to inhibit Na, K–ATPase through PKC-dependent phosphorylation and endocytosis of the α subunit. In hypertensive rats, NHERF-1 expression is reduced, compromising Na, K–ATPase phosphorylation and OUA- sensitive K^+^ transport. The restoration of NHERF-1 expression in these models reverses dopamine-mediated Na, K–ATPase inhibition, suggesting that decreased renal NHERF-1 expression contributes to the imbalance in Na, K–ATPase regulation under hypertensive conditions [134].

Dopamine fails to inhibit Na, K–ATPase in the proximal tubules of hyperinsulinemic obese Zucker rats [212]. This occurrence resulted from a decrease in D1 receptor numbers and a disruption in the coupling of the D1 receptor with G proteins in obese Zucker rats [213,214], and is thought to arise from prolonged cellular exposure to insulin. These findings indicate that hyperinsulinemia may be responsible for disrupting the renal dopaminergic system and diminishing the natriuretic response to external dopamine in individuals with type 2 diabetes, regardless of their hypertension status [215].

The increase in renal levels of Ang II is due both to greater hormonal uptake by AT1 receptors and to renal stimulation dependent on angiotensinogen. The accumulation of this molecule in the lumen of the proximal tubule leads to the local formation of Ang II, which in turn stimulates renal transporters such as the Na^+^/H^+^ exchanger. Local angiotensinogen is associated with the activity of the RAAS, and the increased production of this molecule is reflected in the urine, which is correlated with the RAAS activity index. Hypertensive patients show the increased urinary excretion of angiotensinogen, indicating the positive regulation of the renal RAAS [216]. Elevated angiotensinogen levels can be reduced by treatment with AT1R blockers, suggesting that the activation of this receptor contributes to the progression of hypertension [217].

### 15.2. Renal Hypomagnesemia

Autosomal dominant renal hypomagnesemia is a disorder associated with a mutation in the FXYD2 gene, where a conserved transmembrane glycine is replaced by arginine (121G to A) [218,219]. This mutation leads to the substitution of G41R in the transmembrane domain of the FXYD2 protein, the γ subunit of the Na, K–ATPase. Studies have shown that the G41R mutation disrupts the interaction between Na, K–ATPase and FXYD2 [220,221]. This mutation prevents the translocation of the enzyme to the membrane in *Xenopus laevis* and HeLa cells [218,222].

Experiments conducted with immortalized proximal tubular epithelial cells obtained from the urine of a hypomagnesemic patient showed that the α subunits of Na, K–ATPase and FXYD2 can be positively regulated when exposed to hyperosmolar conditions. However, the absolute expression levels of these proteins were lower in the patient cells than in control cells. Furthermore, the increase in Na, K–ATPase expression is directly proportional to the increase in the enzyme’s maximum activity, without any changes in the apparent affinities for Na^+^, K^+^, or ATP [223].

Hypomagnesemia is also known as a predisposing condition for digitalis toxicity, a serious complication associated with the use of medications such as digoxin. Studies have examined the relationship between digoxin administration and urinary magnesium excretion in patients with atrial fibrillation. The results indicated that the administration of digoxin led to a significant increase in urinary magnesium excretion, supporting the hypothesis that low magnesium levels may contribute to the toxicity of Digoxin, especially in individuals predisposed to this condition, such as those with mutations in the FXYD2 gene [224].

### 15.3. Cancer

In a triple-negative breast cancer cell line, MDA-MB-231, Na, K–ATPase has an important role on Na+-dependent inorganic phosphate transport [225]. OUA also inhibits migration and adhesion of these cell line [225]. The Na, K–ATPase is functionally recognized as a receptor for endogenous molecules, such as CTSs, whose binding triggers complex cellular signaling mechanisms [226]. Thus, the signaling pathways initiated by this enzyme are capable of inducing the proliferation of healthy cells and even inhibiting the proliferation of tumor cells, making Na, K–ATPase a good target in cancer and chemotherapy [226,227,228].

A considerable amount of information in the literature suggests the participation of intracellular junctions such as tight junctions (TJs) and adherens junctions in the development of cancer [229]. The TJs, called zonula occludens, are crucial in establishing and maintaining cell polarity. They exist in the form of macromolecular complexes that consist of different types of proteins, such as transmembrane proteins, cytoskeletal proteins, and signaling molecules, including claudins, occludins, and zonula occludens proteins. Adherent junctions influence the normal development, morphogenesis, and establishment and/or maintenance of different tissues. In this type of junction, E-cadherin is calcium dependent, while catenins are capable of associating with cytoskeletal actins [230]. Na, K–ATPase has been implicated as a strong contributor to the regulation of cell–cell and cell–substrate adhesion in normal and cancerous tissues. Cereijido et al. (2012) established a close relationship between Na, K–ATPase and the loss of cell adhesion, showing that treatment with OUA, a specific inhibitor of Na, K–ATPase, caused the loss of adhesion in MDCK cells [229]. The Na, K–ATPase plays a critical role in the establishment of adherens junctions (AJs) when bound to E-cadherin, the main cytoskeleton protein of AJs [231,232]. The β subunit of Na, K–ATPase has a structure typical of adhesion proteins. Other data indicate that the α and β subunits, which bind and remain bound after their synthesis and insertion into the membrane, are part of the adhesion complex, where there may be β-β interactions between the membranes of contiguous cells may exist. Studies have shown that the overexpression of the Na, K–ATPase beta subunit induces increased adhesion in nonpolarized cells with a colocalization with beta-catenin [233]. Furthermore, the β2 subunit of Na, K–ATPase was indicated as the adhesion molecule on glia, and studies have shown that Chinese hamster ovary fibroblasts transfected with the human β2 subunit become more adhesive and make large aggregates (Figure 5) [234,235]. 

These results suggest that Na, K–ATPase is fundamental for the distribution of TJs and the stabilization of adhesion zones and directly contribute to cell adhesion. This finding was confirmed by studies that revealed a role for OUA in the morphology of adherens junctions [236,237,238]. OUA is capable reducing the expression of the β and α subunits of Na, K–ATPase with signaling cascades dependent on ERK-1 and 2 in a tumor cell model (Caco-2) that does not present caveolae, indicating that the signaling linked to the redistribution of Na, K–ATPase occurred independently of caveolae [238]. Rajasekaran and coworkers reported a reduction in the activity of Na, K–ATPase and also of the pump enzyme expression in an invasive form of human renal carcinoma [239].

The cell adhesion process involves a series of intracellular signaling proteins, such as Src, PKC, calmodulin, and other proteins related to the MAP kinase cascade, which is also makes it dependent on hormonal regulation. Given this discussion, FXYD2 expression appears to be downregulated in some cancerous kidney tissues such as clear cell renal carcinoma [240]. When the sequence of this peptide was analyzed in GenBank (via SAGE tool–Anatomic viewer), FXYD2 mRNA levels were strongly suppressed in liver, kidney, and pancreatic cancer. This alteration may be linked to the phenotypic transformation that occurs during cellular transformation, which culminates in the loss of contact inhibition, unrestrained cell multiplication, and the loss of adhesion observed in cells that acquire metastatic potential, among other changes [241].

Recent research on Na, K–ATPase in cancer has revealed its pivotal role in several cellular processes, including cell adhesion, migration, invasion and intracellular signaling [226,242]. However, methodological limitations pose challenges in fully understanding its role and influence in cancer setting and progression. Thus, the study of Na, K–ATPase in cancer is hindered by the complex regulatory mechanisms of signaling;the differential expression of α, β, and γ isoforms; and the interplay with a lot of cellular processes such as inflammation, the production of cytokines, and the influence of the activation of oncogenes.

Analysis of Na, K–ATPase α- and β-subunit expression in bladder cancer samples shown that decreased enzyme expression levels are linked with the increase in recurrence risk. This suggests that Na, K–ATPase subunit expression might serve as a potential predictor of clinical outcomes [243,244]. Moreover, liver and colorectal cancer metastases exhibit the differential expression of specific isoforms of Na-pump (α1, α3, and β1), with implications for their role in the control of ion gradients and the support of nuclear enzyme functions involved in mitosis control [245]. In this context, the α3 isoform was predominantly located in the cytoplasmic membrane in healthy colon and lung cells. However, the distribution of this isoform was shifted to a predominantly peri-nuclear location in several tumors [246]. It was shown that in liver metastasis the α3 isoform was mainly detected at a peri-nuclear location and was more diffusely expressed across the cytoplasm of tumor metastatic cells. Thus, it is evident that the differential expression and function of Na, K–ATPase isoforms in cancer cells versus normal cells complicate the full comprehension of its role in cancer [226,245].

Future research needs to elucidate the specific contributions of each Na, K–ATPase isoform. In summary, the α3β1 isozyme may have potential as a novel exploratory biomarker for metastases cells. However, further studies still need to be carried out to confirm and expand this apparent involvement.

Voltage-gated sodium channels (VGSCs) and Na, K–ATPase are co-regulated by inflammatory mediators in metastatic breast cancer. This co-regulation is sodium-dependent, and VGSCs are required for the inflammatory mediated effect on Na, K–ATPase [242]. The study of inflammatory mediators like TNFα and prostaglandin E2 on Na, K–ATPase expression in cancer cells shows variability in RNA expression across different cell lines, indicating complex regulatory mechanisms that are not yet fully understood.

The α1 subunit of Na, K–ATPase is intimately associated and regulates the proto-oncogene Src kinase, increasing aerobic glycolysis and tumor growth. The loss of this regulation is associated with increased lactate production and progressive tumor growth, highlighting the Na, K–ATPase/Src kinase complex as a key player as a tumor suppressor [247].

In spite of these challenges, understanding Na, K–ATPase’s roles in cancer progression and treatment resistance has the potential for the development of therapeutic strategies. Further research is crucial to overcome these methodological limitations and fully exploit the Na, K–ATPase as a target in cancer therapy.

The inhibition of Na, K–ATPase in an isoform-selective fashion has been described as a promising strategy for cancer treatment due to its critical role in maintaining cellular functions. Advancements in nanotechnology permit the creation of targeted drug delivery systems that enhance the efficacy and specificity of Na, K–ATPase inhibitors, minimizing side effects and improving therapeutic profiles. DSPE-PEG nanocarrier particles were conjugated with a peptide targeting Na, K–ATPase α1 with the task to deliver Epirubicin (EPI) specifically to breast cancer cells, reducing the size and volume of breast tumors. This targeted approach allows for the slow release of EPI within cancer cells, significantly inhibiting proliferation and migration concurrently reducing systemic collateral effects [248]. In addition, doxorubicin (DOX)-encapsulated nanoparticles with poly(lactic-co-glycolic acid) (PLGA) in composition and fusion with a 13-amino acid peptide targeting Na, K–ATPase exhibited enhanced cellular uptake and antitumor activity in breast cancer models. This strategy improved the main survival rate of tumor-bearing mice and decrease systemic toxicity [249].

Ultra-small vanadate prodrug nanoparticles have been developed to selectively inhibit Na, K–ATPase in cancer cells. This strategy is also known as stimuli-responsive nanoparticles. This kind of nanoparticle is modified with tannic acid and bovine albumin with the goal of reducing systemic toxicity. This nanoparticle is also sensitive to reactive oxygen species generated at the tumor sites. Near-infrared (NIR) photothermal properties further enhance the inhibition of Na, K–ATPase, causing a considerable cancer cell death with a very low impact on healthy tissues [250]. Furthermore, other therapeutic strategies involving vanadate complexes have been described as antitumoral agents. Vanadate derivatives influence lipid peroxidation, induce changes in the cell cycle, and promote the generation of ROS. This inhibitor also affects several signaling pathways, such as AMPK and phosphatase 1B, in breast cancer [251,252].

Some pH-responsive drug delivery nanocarriers were designed to take advantage from the acidic microenvironment of tumors, starting to release drugs only at the tumor site. This strategy increases drug uptake by cancer cells, reducing off-target effects, permitting the resolution of some challenges such as poor tumor selectivity and multidrug resistance [253]. In this context, we can remark that nanomedicine strategies aim to overcome intrinsic and acquired drug resistance by increasing intracellular drug accumulation and selectivity. Multifunctional nanoparticles can also deliver drug combinations employed on synergistic treatments, ameliorating the quality of therapeutic regimens, and overcoming the limitations of conventional chemotherapy [254,255,256].

To date, no unequivocal information is available in the literature regarding the involvement of FXYD2 as an oncogene peptide or a direct participant in the cell adhesion process. However, some studies have shown that FXYD2 is a promising tumor marker for specific types of cancer, such as clear cell ovarian cancer (OCCC), which has a very poor prognosis. In this specific type of cancer, FXYD2 levels can increase approximately 38-fold if we compare microarray data from clear cell cancer with those from ovarian and endometrial control cells [257]. Hsu and coworkers (2016) reported significantly higher FXYD2 expression in advanced-stage OCCC than in early-stage OCCC, which was positively correlated with the patient prognosis [258].

Cisplatin and other platinum-containing drugs have played crucial roles in anticancer treatments for more than 30 years. However, cisplatin treatment can cause serious side effects, such as myelo suppression, nausea, ototoxicity, nephrotoxicity, and cellular resistance. Furthermore, since the last decade, cardiotonic steroids, particularly digoxin, have been tested for their anticancer effects, with several pivotal results reported (for a review, see Nabil et al., 2024; Wang et al., 2021) [259,260]. Therefore, Pereira et al. (2018) showed that combined treatment of HeLa cells with cisplatin and digoxin improved the cytotoxic effects and decreased the side effects of cisplatin. The interaction between cisplatin and digoxin had a synergistic effect on cervical cancer cells and a significantly positive cytotoxic and antiproliferative effect on this cell line compared with the control treatments and cisplatin alone [227]. Although a decrease in the expression of the α1 subunit of Na, K–ATPase was observed in total extracts, its expression remained unchanged in the membrane, as did Na, K–ATPase activity. The antiproliferative effect of the synergistic treatment appears to depend on the activation of Src kinase, indicating the possible involvement of the Src-EGFR-ERK1/2 pathway in the antitumor effect (Figure 6). The inhibition of ERK1/2 caused the same synergism with 1 μM cisplatin as that observed with 1 nM digoxin plus 1 μM cisplatin but not with 1 nM digoxin alone. Pretreatment with PP2 (a Src inhibitor) during the combination treatment abolished the synergistic inhibitory effect on proliferative activity. Cisplatin and digoxin are already used in current clinical treatment protocols; thus, these studies open possibilities for future clinical trials employing combined treatments to improve antitumoral chemotherapy with lower toxicity and side effects. These findings become even more relevant in light of the work of Nie et al., 2020, who elegantly demonstrated that Na, K–ATPase unequivocally forms functional complexes with Src kinase and E-cadherin under native conditions and after crosslinking Cys residues using BMH and DTME [261]. In this context, the transition of the pump enzyme from the E1 to the E2 form can affect the stability of these complexes. These interactions are decisive for the functioning of the Na^+^ pump as a receptor for hormones and CTSs.

Na, K–ATPase is the main receptor for cardiotonic steroids such as digoxin, which is often associated with intracellular cisplatin accumulation. Several studies have described the effects of cisplatin on Na, K–ATPase [262,263,264,265]. Compared with untreated cells, cell lines that are resistant to cisplatin exhibit lower Na, K–ATPase expression. Na, K–ATPase expression is directly related to the cellular susceptibility to cisplatin [266]. As shown in a previous study [267], cisplatin-sensitive cell lines highly express the Na, K–ATPase α1 subunit after cisplatin treatment, which can be correlated with drug sensitivity.

Oxaliplatin and carboplatin (other platinum-based chemotherapy formulations) have much lower nephrotoxicity than cisplatin, and these two drugs do not inhibit Na, K–ATPase [268,269]. Furthermore, in 2018, Pereira et al. revealed that the sensitizing effect of digoxin is not related to the activity of Na, K–ATPase in HeLa cells; instead, they postulated that the effect is linked to the intracellular signaling triggered by this combination of drugs [227].

Another CTS of natural origin, glucoevatromonoside (GEV), also has great potential as an anticancer treatment because it also synergizes with cisplatin and paclitaxel, exerting antiproliferative effects on H460 lung cancer cells [228]. In this context, in addition to the antiproliferative effect, reductions in invasion and migration were also reported, indicating a potential reduction in the aggressiveness of this tumor lineage after the combined treatment of GEV with certain chemotherapy drugs. Other studies have shown that FXYD2 is already expressed in several types of cancer, and its expression is altered in many of them. For example, Jin et al. (2021) reported that FXYD2 expression was upregulated in colorectal cancer tissues compared with normal tissues [270]. Similarly, a study by Li et al. (2022) indicated that FXYD2 was overexpressed in hepatocellular carcinoma tissues compared with normal liver tissues [271]. These findings suggest that FXYD2 may play a potentially important role in the development of cancer.

The mechanisms by which FXYD2 promotes cancer development are not fully understood. However, some studies have suggested that FXYD2 may contribute to cancer progression by regulating signaling pathways involved in cell proliferation and survival. For example, a study reported that FXYD2 promoted the activation of the PI3K/AKT signaling pathway [240], which is known to be involved in the survival and proliferation of cancer cells. Similarly, a study by Frank et al. (2024) found that FXYD2 was one of four main genes involved in the progression of cholangiocarcinoma promoting cell proliferation and migration in cholangiocarcinoma, since the overexpression of FXYD2 was implicated in dysplasia and the poor prognosis of patients from which organoids excised from gallbladder epithelial cells carcinoma were analyzed [272].

FXYD3 (antigen-8 of mammary tumor) is known to exert protection of the β1 subunit against glutathionylation, an oxidative modification that destabilizes the α-β heterodimer and inhibits Na, K–ATPase activity. A specific cysteine (Cys) residue of FXYD proteins seems to be crucial for such protection. One of the FXYD proteins, FXYD3, confers chemotherapy resistance when it is overexpressed in cancer cells. Chemotherapeutic compounds, such as doxorubicin, can induce oxidative stress, and Liu et al., 2022, showed that the suppression of FXYD3 with siRNA in pancreatic and breast cancer cells, which strongly express FXYD3, increased doxorubicin-induced cytotoxicity. This process restored the sensitivity of these tumor cells to one of the most used chemotherapies [111].

Bioinformatic analysis tools were determinant to evaluate the role of a high expression of KDM3A (lysine demethylase 3A) and DCLK1 (doublecortin-like kinase 1) and reduced expression FXYD3 in lung cancer. In the case of DCLK1, this article demonstrated its role in the FXYD3 suppression [273]. A reduced expression of FXYD3 was reported in lung cancer cells, in which its inactivation was identified as a key player in lung cancer progression [274]. In lung tumors, the increasing levels of DCLK1, promotes the proliferation and metastasis of lung cancer cells through the downregulation of FXYD3 [273].

Finally, the compilation of the latest data on the role of these peptides in cancer development and the underlying biochemistry indicate that FXYD2 (ATP1G1) and FXYD3 (MAT-8) regulate the activity of the Na, K–ATPase pump and may play a role in regulating cell proliferation, migration, and invasion, key processes in cancer growth and metastasis (Table 1).

The current literature suggests that FXYD2 may play a role in cancer development and progression by regulating ion transport and signaling pathways involved in cell proliferation and survival. Changes in FXYD2 expression and genetic variants in FXYD2 were shown to be associated with an increased cancer risk in several studies.

## 16. Future Directions on Na, K–ATPase and Cancer Research

Some challenges still surround the future of Na, K–ATPase-related cancer research, in which the optimization and precise targeting associated with the better control of drug release might be viewed as the main promise of nanomedicine. If this goal were fully achieved, it would increase the specificity and toxicity of existing nanoparticle-based systems, improving therapeutic efficacy and clinical outcomes [256]. In spite of the current intrinsic potential of nanocarriers, only a few real nanomedicine drugs have received FDA approval for use as cancer treatments. Continued research and clinical trials are necessary to validate the safety and effectiveness of novel nanocarriers and to facilitate their translation into clinical practice [255].

Na, K–ATPase is very well known as a target for cardiotonic steroids, which shows promise in overcoming multi-drug resistance in cancer cells. However, the development of resistance by these compounds and their precise mechanisms of action require further investigation [275]. Combined treatments employing the synergistic effects of CTS and chemotherapies, in association with multidrug deliverance systems, would be very useful in future cancer research [227]. Further research and clinical validation are essential for realizing the full potential of these innovative therapeutic strategies.

## 17. Conclusions

In summary, this review article provides an overview of Na, K–ATPaseregulation, integrating molecular biology and biochemistry with clinical relevance. The article effectively details how phosphorylation by various kinases serves as a critical regulatory mechanism for Na, K–ATPaseactivity. This understanding is essential for grasping how cellular ion homeostasis is maintained and how disruptions can lead to pathological states. A significant focus is placed on the role of hormones and FXYD proteins and the complexity of their interactions. It illustrates how hormones can exert both stimulatory and inhibitory effects depending on their concentrations and the cellular context. The inclusion of FXYD proteins, particularly FXYD2, adds depth to the discussion. The ultimate boom in the recent literature regarding FXYD proteins just reinforces this point of view. These proteins are shown to modulate the activity and stability of Na, K–ATPase, linking them to various physiological and pathological conditions. In this regard, this article connects the biochemical mechanisms of Na, K–ATPase to clinical outcomes, particularly for diseases such as hypertension and cancer. By illustrating how the dysregulation of this enzyme contributes to disease pathology, this article underscores the potential for targeted therapeutic strategies aimed at restoring normal Na, K–ATPasefunction, thus paving the way for advancements in health and the treatment of diseases linked with Na, K–ATPase dysregulation.

## Figures and Tables

**Figure 1 ijms-25-13398-f001:**
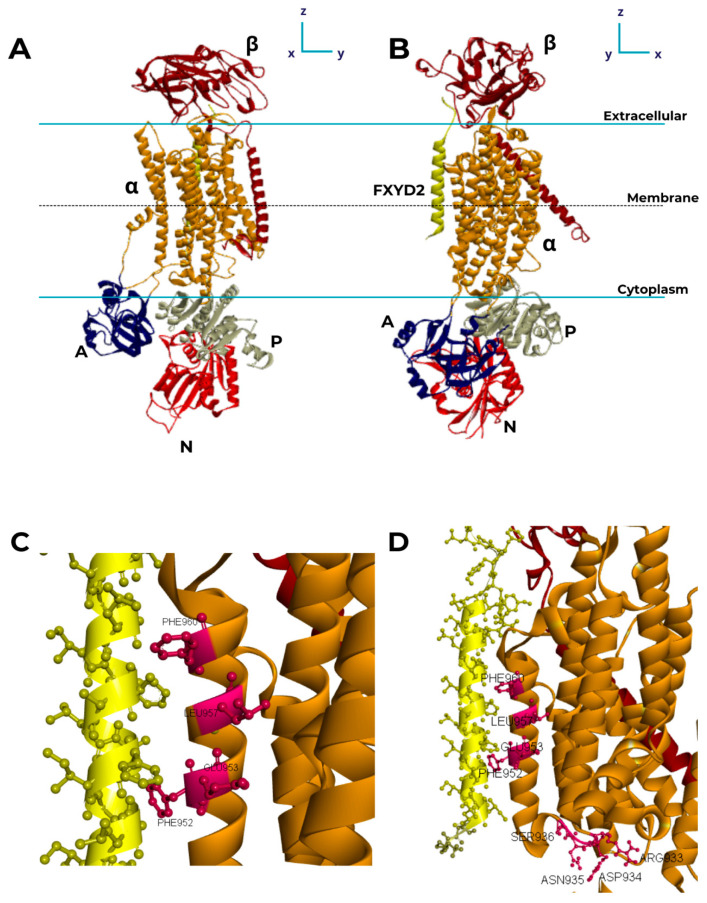
**Crystal structure of the Na, K–ATPase in the E1.Mg^2+^ state**. Panels (**A**,**B**) show the Na, K–ATPase from pig kidney viewed from two orthogonal directions. The enzyme consists of a catalytic α-subunit (orange), a glycosylated β-subunit (maroon), and a regulatory FXYD protein, specifically FXYD2 (yellow), located behind the α-subunit. The α-subunit comprises three well-defined cytoplasmic domains (A-blue, N-red, and P-gray) and 10 transmembrane helices (M1–M10). The helices are not labeled with numbers. Panel (**C**) highlights residues (pink) in the M9 region of the α subunit that interact with the corresponding FXYD peptides and are important for forming a stable complex [13]. Panel (**D**) shows the RRNS motif of the Na, K–ATPase α1 isoform, where Ser936 is phosphorylated by PKA [13]. Protein Data Bank (PDB) ID: 8JBL.

**Figure 2 ijms-25-13398-f002:**
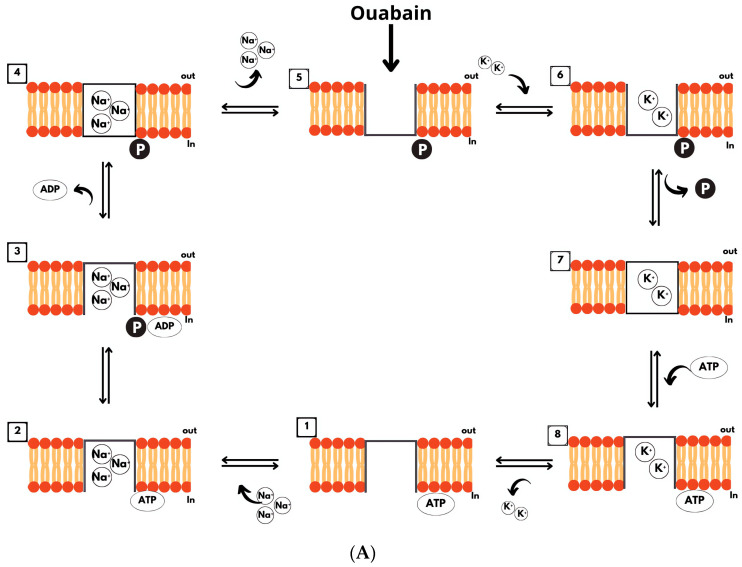
**Scheme of catalytic cycle of Na, K–ATPase**. Panel (**A**,**B**) represent the simplified Post–Albers model [11] that outlines the process of ATP hydrolysis and ion transport by the Na, K–ATPase. This enzyme alternates between two conformations, E1 and E2. In the forward cycle (clockwise), the Na, K–ATPase first binds intracellular Na^+^ and MgATP with high affinity, forming the Na3E1ATP complex (Mg ions are not shown). The γ-phosphate of ATP is then transferred to the enzyme, and Na^+^ ions become occluded (represented by parentheses). The resulting (Na3)E1-P^•^ADP complex has a high-energy phosphate bond, making the reaction reversible. After ADP is released, Na^+^ ions are deoccluded and expelled into the extracellular space following or alongside the enzyme’s conformational shift in the enzyme to E2-P. This E2-P conformation also serves as the binding site for OUA, a well-known inhibitor of Na, K–ATPase. Extracellular K^+^ then binds to E2-P, promoting Pi release and K^+^ occlusion as they travel to the cytosol (protons, which are thought to bind to the “third Na^+^ site” with two K^+^ ions, are not shown). ATP, acting with low apparent affinity, accelerates K^+^ deocclusion and intracellular release. The enzyme then shifts back from E2 to E1 and is ready to begin the cycle again.

**Figure 3 ijms-25-13398-f003:**
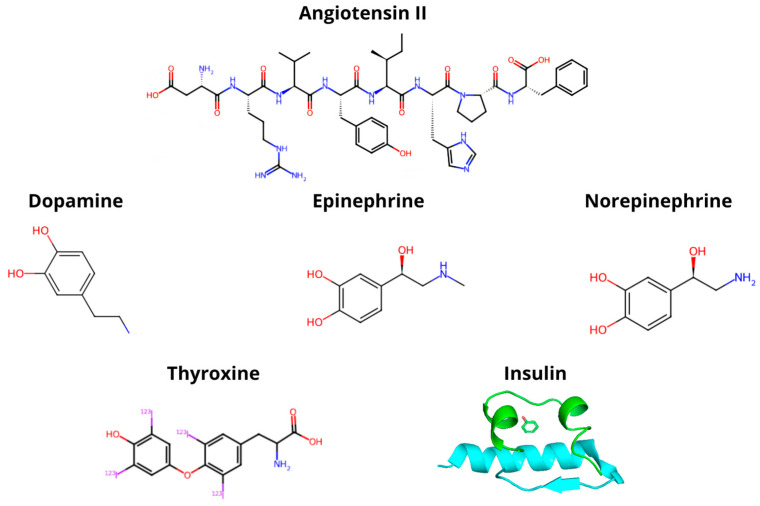
**Representation of the chemical structures of hormones that regulate Na, K–ATPase**. The structures are represented as follows: Angiotensin II in green, dopamine in pink, epinephrine in yellow, norepinephrine in salmon, thyroxine in purple, and the three-dimensional structure of insulin (PDB: 1WAV). Oxygen atoms are shown in red, nitrogen atoms in blue, and iodine-123 in purple. The small molecule structures were obtained from ChemSpider, with CSIDs 150504, 661, 5611, 388394, and 64880242, respectively.

**Figure 4 ijms-25-13398-f004:**
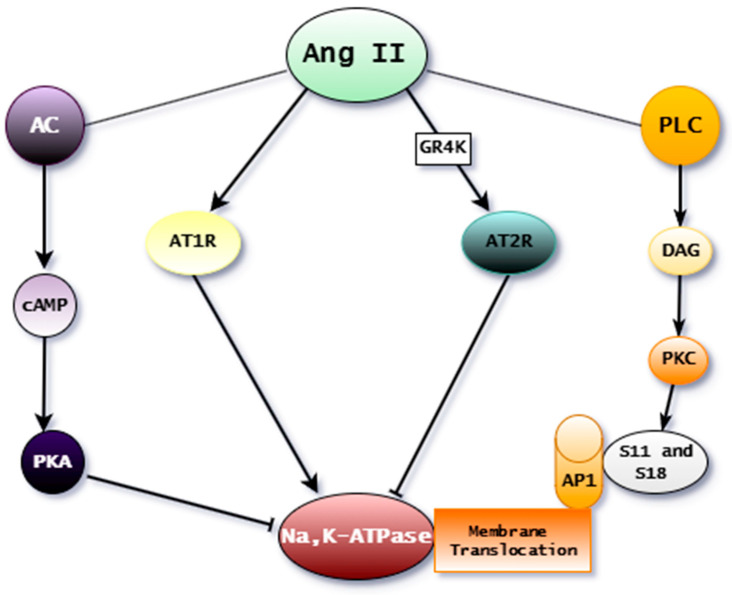
**Ang II induces multiple signaling pathways that regulate Na, K–ATPase activity**. In the adenylate cyclase–cAMP–PKA pathway, the phosphorylation of the α subunit by PKA inhibits Na, K–ATPase activity. Stimulatory effect of Ang II via the AT1R. The GRK4 increased phosphorylation of AT2R is associated with the inhibition of Na, K–ATPase. A signaling pathway involving PKC and the interaction of Na, K–ATPase with the adaptor protein 1 (AP1) recruits Na, K–ATPase molecules to the plasma membrane.

**Figure 5 ijms-25-13398-f005:**
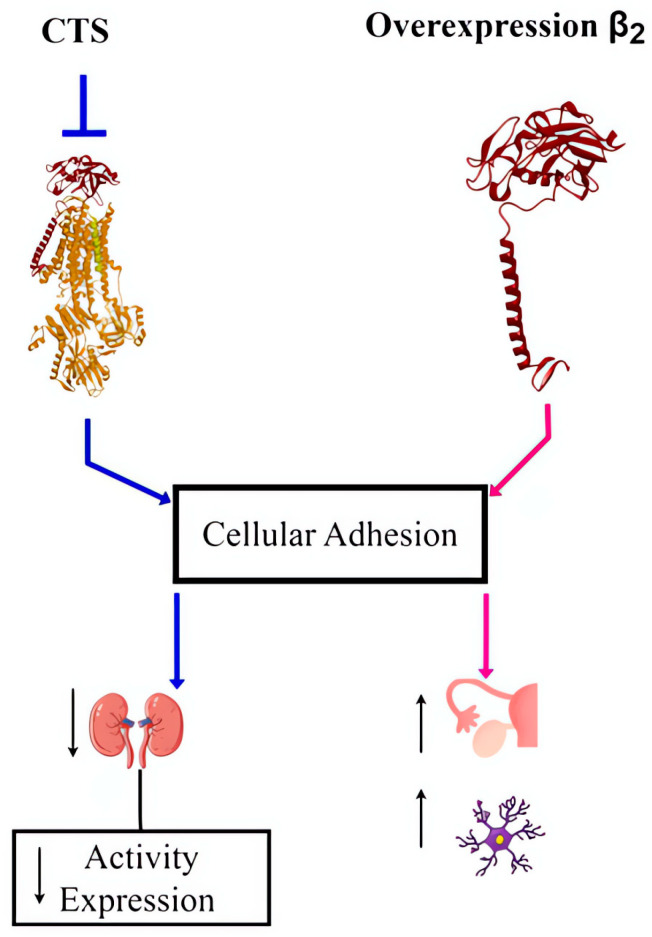
**Schematic diagram of the role of Na, K–ATPase in cell adhesion in different cancer types**. Blue line shows that Na^+^ pump inhibition by CTS reduces cell adhesion in renal cells, leading to decreased expression and enzymatic activity, which is linked to cancer progression. Pink line shows that the overexpression β_2_ isoform increased cellular adhesion on glia and ovary, arresting cancer progression.

**Figure 6 ijms-25-13398-f006:**
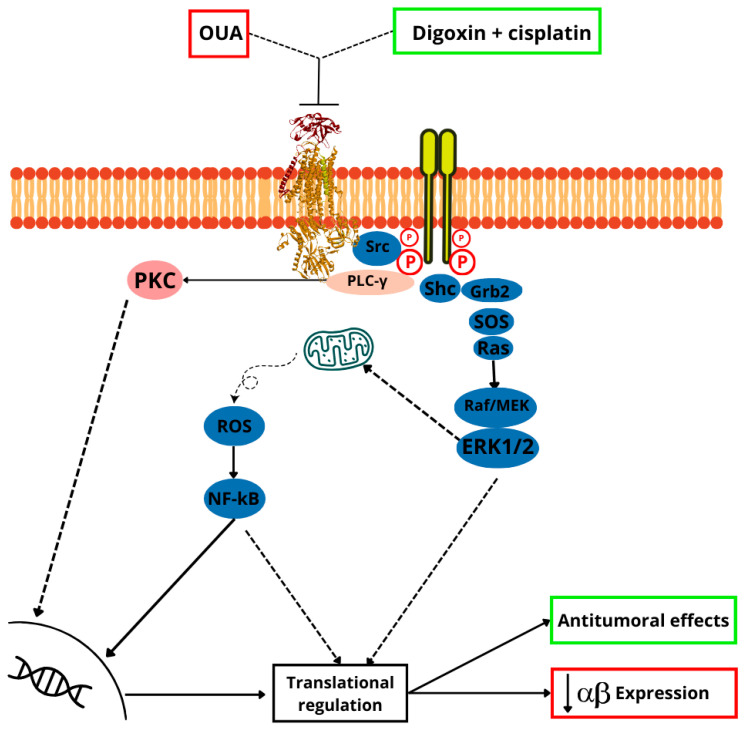
**Signaling pathways of Na, K–ATPase and their consequences in cancer**. The inhibition of Na, K–ATPase by ouabain (red boxes) and the combined treatment with digoxin and cisplatin (green boxes) activate the Src/EGFR pathway, which leads to the activation of ERK and increased ROS production in mitochondria. This activates NF-κB, resulting in transcriptional regulation. EGFR phosphorylated activates the PKC pathway. Ouabain decreases the expression of alpha and beta subunits, while the combined treatment with digoxin and cisplatin has antitumoral effects. Solid arrows indicate experimentally supported events induced by inhibitors mentioned, while broken arrows indicate events with limited or indirect support.

**Table 1 ijms-25-13398-t001:** Differential expression of Na^+^-pump enzyme isoforms and prognosis of several types of cancer.

Cancer	Isoform	Expression Level	Prognosis
Colorectal	α1β1	Decreased	Unknown (Bechmann et al., 2016) [245]
Colorectal	α3β1 (perinuclear)	Increased	Poor (Yang et al., 2014) [246]
Hepatocellular carcinoma	α3β1 (perinuclear)	Increased	Poor (Yang et al., 2014) [246]
Cholangiocarcinoma	FXYD2	Increased	Poor (Frank et al., 2024) [272]
Hepatocellular carcinoma	FXYD2	Much increased	Poor (Li et al., 2022) [271]
Pancreatic	FXYD2	Decreased	Poor (Zhang et al., 2022) [240]
Clear cell renal carcinoma	FXYD2	Decreased	Poor (Zhang et al., 2022) [240]
Clear cell renal carcinoma	FXYD2	Much increased	Better (Zhang et al., 2022) [240]
Colorectal	FXYD2	Increased	Poor (Jin et al., 2021) [270]
Clear cell ovarian cancer	FXYD2	Much Increased (Schwartz et al., 2002) [257]	Poor (Hsu et al., 2016) [258]
Breast	FXYD3	Increased	Poor (Liu et al., 2022; Li et al., 2023) [111,250]
Lung	FXYD3	Normal	Better (Liu et al., 2021) [273]

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
