# Peer review of "Molecular Basis of Na, K–ATPase Regulation of Diseases: Hormone and FXYD2 Interactions"

_ijms, 2024, doi:10.3390/ijms252413398_

Round 1

Reviewer 1 Report

Comments and Suggestions for Authors

The author’s purpose of the review paper about “ Molecular basis of Na,K‒ATPase regulation of diseases: Hor-2 mone and FXYD2 interactions.” is very interesting also for related research fields.  The paper is globally well written and easy to understand. Suggestions:

I would also like to suggest, that besides the specific ATPase inhibitor oubain the authors could eventually also refer, and also to turn the paper with a broader  interest, to the specific specific inhibitor vanadate, the one described at  the Nobel prize at 1997 when the discover of the molecular mechanisms of the P-type ATPases, once it was referred that these ATPases were specifically inhibited by vanadate at the conformation E2-P.

More recently, besides vanadate, decavanadate  have been referred to be a more potent inhibitor of these P-type ATPases and also described to be used against cancer as well as others pathologies (see for example, De Sousa Coelho, Pharmaceuticals, 2024, 17, 12; Aureliano et al CCR 2021 and CCR 2022) not excluding the possibility of these POMs to be specific a regulators of these ATPases in disease. Moreover, it was recently described for both PMCA and SERCA as well as for purinergic ionotropic and metabotropic receptors (see for example, Poejo et al JIB 2024 and Aureliano JIB 2022) can be modulated by these inorganics compounds in several physiological processes and in disease. That is, besides platin compounds, others inorgabics compounds referred since the beggining to be specific inhibitors of these ATPases (and not just cisplatin, that has no similar inhibitory potential ability as vanadate) could favor the review paper.

Author Response

Comment 1: The author’s purpose of the review paper about “ Molecular basis of Na,K‒ATPase regulation of diseases: Hor-2 mone and FXYD2 interactions.” is very interesting also for related research fields.  The paper is globally well written and easy to understand. Suggestions:

Comment 2: I would also like to suggest, that besides the specific ATPase inhibitor oubain the authors could eventually also refer, and also to turn the paper with a broader  interest, to the specific specific inhibitor vanadate, the one described at  the Nobel prize at 1997 when the discover of the molecular mechanisms of the P-type ATPases, once it was referred that these ATPases were specifically inhibited by vanadate at the conformation E2-P.

Answer 1/2:

Many thanks for your prestigious comments. A brief description regarding the properties of vanadate and decavanadate as selective P-type ATPase inhibitors was added to the text of the manuscript (Page 14, section “Vanadate”, 2th paragraph).

Vanadate

Vanadate is a phosphate transition state analog that stabilizes an E2 conformation of the Na, K‒ATPase, mimicking the E2-P or E2• Pi conformation, it acts as a blocker at the phosphate discharge site [89,90]. Thr214→Ala mutations of the TGES conserved motif of the phosphorylation site displayed a conspicuous 151-fold reduction of the apparent vanadate affinity during catalytic turnover in the ATPase assay [91].  Vanadate is one of the most famous and studied P‒type ATPase inhibitors. However, even after four decades since this discovery, several POMs (Polyoxometalates) still have been considered as very promising drugs on the treatment of several diseases including cancer. Any disease in which Na, K‒ATPase, SERCA, or H,K‒ATPase can be ascribed as the main target would have a POM as a drug choice for potential treatment [91].  The decavanadate is considered a potent mixed-mode inhibitor of P-type ATPases that alters the kinetics of Na, K‒ATPase from rat synaptosomes (IC50=470 nM). It can inhibit the enzyme decreasing the maximum velocity and the apparent affinity for ATP, thus increasing Km for substrate [92].

The decavanadate is considered a potent mixed-mode inhibitor of P-type ATPases that alters the kinetics of Na, K‒ATPase from rat synaptosomes (IC50=470 nM). It can inhibit the enzyme decreasing the maximum velocity and the apparent affinity for ATP, thus increasing Km for substrate [93].”

Comment 3: More recently, besides vanadate, decavanadate  have been referred to be a more potent inhibitor of these P-type ATPases and also described to be used against cancer as well as others pathologies (see for example, De Sousa Coelho, Pharmaceuticals, 2024, 17, 12; Aureliano et al CCR 2021 and CCR 2022) not excluding the possibility of these POMs to be specific a regulators of these ATPases in disease. Moreover, it was recently described for both PMCA and SERCA as well as for purinergic ionotropic and metabotropic receptors (see for example, Poejo et al JIB 2024 and Aureliano JIB 2022) can be modulated by these inorganics compounds in several physiological processes and in disease. That is, besides platin compounds, others inorgabics compounds referred since the beggining to be specific inhibitors of these ATPases (and not just cisplatin, that has no similar inhibitory potential ability as vanadate) could favor the review paper.

Answer 3:

We totally agree with this comment and we inserted some of your reference suggestions and some additional references at the text of the manuscript (Page 34, 2th paragraph). This paragraph includes some recent advances as the use of stimuli-oriented nanoparticles carrying vanadate to specifically deliver the inhibitor and kill cancer cells. This addendum goes as follows:

“          Ultra-small vanadate prodrug nanoparticles have been developed to selectively inhibit Na, K‒ATPase in cancer cells. This strategy is also known as stimuli-responsive nanoparticles. This kind of nanoparticle is modified with tannic acid and bovine albumin with the goal to reduce systemic toxicity. This nanoparticle is also sensitive to reactive oxygen species generated at the tumor sites. Near-infrared (NIR) photothermal properties further enhance the inhibition of Na, K‒ATPase, causing a considerable cancer cell death with a very low impact on healthy tissues [255].  Furthermore, other therapeutic strategies involving vanadate complexes have been described as antitumoral agents. Vanadate derivatives influences lipid peroxidation, induces changes in the cell cycle, and promotes the generation of ROS. This inhibitor also affects several signaling pathways, such as AMPK and phosphatase 1B, in breast cancer [256,257].”

Reviewer 2 Report

Comments and Suggestions for Authors

The review article proposed to me for review is interesting and useful in the subject matter it covers. This review explores Na, K‒ATPase regulation by hormones and signaling pathways, focusing on FXYD proteins (FXYD1 and FXYD2) and their phosphorylation. It examines the effects of CTS, angiotensin II, dopamine, insulin, and catecholamines, linking these mechanisms to diseases like hypertension, renal hypomagnesemia, and cancer, highlighting Na, K‒ATPase as a key therapeutic target.

The abstract feels very simple, please make it more informative. The abstract ends with no comma. Please add.

Please add few more key words, for example: Ion transport, hormonal regulation, signal transduction, protein kinases. Etc.

The full text needs a punctual checking. Please correct it! There are missing spaces everywhere.

The article seems unfinished and left in the middle of nowhere. Please critically comment on the information to make the review article worthwhile. Include a conclusion and only state the key aspects of your review article.

The references in the text should be only in numbers. Please revise.

I have a few questions - which comments the authors can include in the manuscript:

How do the two splice variants of FXYD2 (γa and γb) differ in their regulatory effects on Na, K‒ATPase activity, and what implications might these differences have for tissue-specific functions?

In the context of diseases mentioned (e.g., hypertension, renal hypomagnesemia, cancer), how might alterations in Na, K‒ATPase or FXYD2 expression contribute to disease progression or severity?

Could you elaborate on how oxidative stress influences Na, K‒ATPase activity and its interaction with FXYD proteins? Are there specific pathways involved that could be targeted therapeutically?

Please represent the chemical structures in Figure 3 with simple skeletal structural formulas.

Based on the above I suggest the article be reconsidered after major revisions.  

Comments on the Quality of English Language

The English language is fine. Minor errors detected. Please check all the text for spelling, punctual, and grammar errors. 

Author Response

Comment 1: The review article proposed to me for review is interesting and useful in the subject matter it covers. This review explores Na, K‒ATPase regulation by hormones and signaling pathways, focusing on FXYD proteins (FXYD1 and FXYD2) and their phosphorylation. It examines the effects of CTS, angiotensin II, dopamine, insulin, and catecholamines, linking these mechanisms to diseases like hypertension, renal hypomagnesemia, and cancer, highlighting Na, K‒ATPase as a key therapeutic target.

Comment 2: The abstract feels very simple, please make it more informative. The abstract ends with no comma. Please add.

Answer 1/2: Many thanks for your prestigious comments, we appreciated your opinion about our manuscript. We agree with your suggestion regarding the abstract, and it was revised and expanded to gather more clarity. The revised abstract goes as follows:

“The Na, K‒ATPase generates an asymmetric ion gradient that supports multiple cellular functions, including the control of cellular volume, neuronal excitability, secondary ionic transport, and the movement of molecules like amino acids and glucose. The intracellular and extracellular levels of Na⁺ and K⁺ ions are the classical local regulators of the enzyme's activity. Additionally, the regulation of Na, K‒ATPase is a complex process that occurs at multiple levels, encompassing its total cellular content, subcellular distribution, and intrinsic activity. In this context, the enzyme serves as a regulatory target for hormones, either through direct actions or via signaling cascades triggered by hormone receptors. Notably, FXYDs small transmembrane proteins regulators of Na, K‒ATPase serve as intermediaries linking hormonal signaling to enzymatic regulation at various levels. Specifically, members of the FXYD family, particularly FXYD1 and FXYD2, are that undergo phosphorylation by kinases activated through hormone receptor signaling, which subsequently influences their modulation of Na, K‒ATPase activity This review describes the effects of FXYD2, cardiotonic steroid signaling, and hormones such as angiotensin II, dopamine, insulin, catecholamines on the regulation of Na, K‒ATPase. Furthermore, this review highlights the implications of Na, K‒ATPase in diseases such as hypertension, renal hypomagnesemia, and cancer.”

Comment 3: Please add few more key words, for example: Ion transport, hormonal regulation, signal transduction, protein kinases. Etc.

Answer 3: Thanks for your comment key word list were promptly expanded: the revised list goes as follows:

“Keywords: Na, K‒ATPase; FXYD2; ion transport; hormonal regulation; signal transduction; receptors; protein kinases; diseases; hypertension; renal hypomagnesemia; cancer.”

Comment 4: The full text needs a punctual checking. Please correct it! There are missing spaces everywhere.

Answer 4: The text was checked and corrected throughout the revised version.

Comment 5: The article seems unfinished and left in the middle of nowhere. Please critically comment on the information to make the review article worthwhile. Include a conclusion and only state the key aspects of your review article.

Answer 5: We think that the manuscript was abruptly ended, so your original comment was alright. Thus, we introduce a brief general conclusion regarding the entire review to address this query. This amendment goes as follows:

“In summary, this review article provides an overview of Na, K‒ATPase regulation, integrating molecular biology and biochemestry with clinical relevance. The article effectively details how phosphorylation by various kinases serves as a critical regulatory mechanism for Na, K‒ATPase activity. This understanding is essential for grasping how cellular ion homeostasis is maintained and how disruptions can lead to pathological states. A significant focus is placed on the role of hormones and FXYD proteins and the complexity of their interactions. It illustrates how hormones can exert both stimulatory and inhibitory effects depending on their concentrations and the cellular context. The inclusion of FXYD proteins, particularly FXYD2, adds depth to the discussion. The ultimate boom on the recent literature regarding FXYD proteins just reinforces this point of view. These proteins are shown to modulate the activity and stability of Na, K‒ATPase, linking them to various physiological and pathological conditions. In this regard, the article connects the biochemical mechanisms of Na, K‒ATPase to clinical outcomes, particularly in diseases such as hypertension and cancer. By illustrating how dysregulation of this enzyme contributes to disease pathology, the article underscores the potential for targeted therapeutic strategies aimed at restoring normal Na, K‒ATPase function. Thus paving the way for advancements in health and the treatment of diseases linked with Na, K-ATPase dysregulation.”

Comment 6:  The references in the text should be only in numbers. Please revise.

Answer 6: The specific case in which this mistake was evident (page 38, line 01 revised manuscript) was promptly corrected, many thanks for your observation. There are many cases in which the name of authors was mentioned on the flux of the text and it is remain unchanged because we don´t want to change this style of writing (all these citations are adequately numbered).

Comment 7: How do the two splice variants of FXYD2 (γa and γb) differ in their regulatory effects on Na, K‒ATPase activity, and what implications might these differences have for tissue-specific functions?

Answer 7: Many thanks for the smart comment, we inserted one text regarding the aminoacid sequence description of FXYD2 variants and also about its differential properties and involvement. This small section goes as follows and it was placed in page 9 of the revised manuscript:

“FXYD2, also called the γ subunit, was originally thought to be the third component of the Na, K‒ATPase complex. The γ subunit exists in two splice variants, γa (7.184 kDa) and γb (7.338 kDa), which are found in the kidneys and pancreas [44–46]. Research by Kuster et al. (2000) revealed that these variants differ in seven N-terminal residues, γa, TELSANH, are substituted by Ac-MDRWYL in γb, but the rest of the sequence is the same [47]. The localization of FXYD2a and FXYD2b variants was described in the literature along with nephron [48–53]. Both forms are predominantly expressed in the thick ascending limb and similarly reduce the apparent sodium affinity of Na, K‒ATPase [38,45]. The FXYD2a and FXYD2b variants appear to undergo different posttranslational modifications that affect enzyme kinetics differently, although the physiological purpose of having two isoforms remains unclear [9,44]. The kinectic effects of the two variants are very similar, the works suggest that both decresead the apparent sodium affinity [54]. Consequently, it causes a reduction of Na, K‒ATPase activity [46,55]. The expression of FXYD2 is restricted to the kidney and pancreas, your posttranslational modifications and effects on Na, K‒ATPase are poorly known. The localization of FXYD2a and FXYD2b variants was described in the literature along with nephron [48–53]. Some studies demonstrated that mutation in FXYD2b affects the regulation of the cell surface and cell growth [56]. The physiological and pathological role of FXYD2a and FXYD2b in the kidney remains unknown. However, in the pancreas, research genomic-based proposed FXYD2a as a novel beta cell-specific biomarker. Histological examinations of pancreatic sections from individuals with type-1 diabetes, as well as sections from streptozotocin-treated monkeys, revealed a correlation between the loss of FXYD2a expression and a reduction in insulin-positive cells. Consequently, FXYD2a has been proposed as a potential biomarker, with its application offering a valuable tool for monitoring cellular mass under various conditions, such as type 1 diabetes [57].”

Comment 8: In the context of diseases mentioned (e.g., hypertension, renal hypomagnesemia, cancer), how might alterations in Na, K‒ATPase or FXYD2 expression contribute to disease progression or severity?

Answer 8:  To answer this question, we remark that Referee 3 also made the same suggestion. Thus we revised throughout the text of hypertension (page 27, 4th paragraph, line 6) and cancer (page 32 till the end of the cancer section) to address both referee´s comments. As we made a lot of insertions we would ask the referee to read these sections carefully (all changes are marked in red). A newly designed table was performed to discuss the contribution of many Na, K‒ATPase isoforms, or even combinations of them, during cancer setting and progression. Moreover, two originally designed schemes were inserted in the cancer section text (figure 5, page 31, and figure 6, page 36).

Comment 9: Could you elaborate on how oxidative stress influences Na, K‒ATPase activity and its interaction with FXYD proteins? Are there specific pathways involved that could be targeted therapeutically?

Answer 9: Many thanks for your interest on this subject. To answer or address your comment we again inserted a subsection on the review text (page 15 of revised manuscript) regarding this matter and it goes as follows:

“Oxidative stress

The oxidative stress is related to an increase in the generation of reactive oxygen species (ROS), ROS causes several effects on the Na, K‒ATPase and this has been extensively described in the literature. Oxidative stress causes many modifications in proteins like s-glutathionylation, s-nitrosylation and carbonylation, among others.  Elevates levels of ROS induce oxidative modification of the Na, K‒ATPase α and β subunits alongside FXYD proteins [94–98]. These oxidative modifications caused by superoxide, nitrite peroxide, S-nitrosoglutathione (GSNO), and other free radicals donors can irreversible inhibit Na, K‒ATPase through phospholipid peroxidation at vicinity of the catalytic site [99]. The lipid peroxidation impaired Na, K‒ATPase through modifications of α1 subunit in kidney exacerbating the dissipation of monovalent cation gradients [100,101]. Coronary artery disease (CAD) patients had higher levels membrane lipid peroxidation that was negatively correlated with the decreased Na, K‒ATPase activity. On the other hand, it is positively correlated with the severity of CAD [102].

Hyperglycemia induces an increase in endothelial superoxide that inhibits the stimulatory effect of nitric oxide (NO) on vascular Na, K‒ATPase activity [103]. Nitrite peroxide is the ROS product of the reaction of NO and superoxide. Besides, it is a potent inhibitor of Na, K‒ATPase activity, and it can induce amino acid modifications to the pump [104,105].

Notably, the S-glutathionylation of the β1 subunit, along with the associated inhibition of enzyme activity, can be dynamically reversed by FXYD proteins [95]. Research has shown that ROS are pivotal in the Na, K‒ATPase signaling pathway. ROS induces modifications in the α1 subunit of Na, K‒ATPase, leading to the activation of Src's tyrosine kinase activity. This activation initiates a signaling cascade that amplifies ROS production and modulates other pathways, including the Ras/Raf/MEK/ERK1/2 cascade, mitochondrial ROS generation, PLC/PIP2/inositol triphosphate, PLC/PIP2/diacylglycerol/protein kinase C, and PI3K/AP2/clathrin-mediated endocytosis, along with their downstream effects [106]. Consequently, the Na, K‒ATPase signaling cascade appears to function as a feed-forward mechanism that amplifies ROS, with circulating cardiotonic steroids modulating its intensity. Evidence from cellular and animal disease models indicates that this amplification mechanism is activated under oxidative stress conditions, such as obesity/metabolic syndrome, chronic kidney disease, cardiovascular disease, cancer and other related disorders commonly linked to oxidative imbalance. Therefore, it presents a promising therapeutic target for clinical intervention [107,108].

The relationship between oxidative stress and the enzymatic interaction with FXYD proteins varies depending on the isoform expressed and the cell type involved in different pathologies. FXYD1 has been identified as playing a crucial protective role in vascular health, particularly by shielding the Na, K‒ATPase from oxidative inhibition. Silencing FXYD1 in human umbilical vein endothelial cells led to a 50% reduction in nitric oxide (NO) production and a twofold increase in superoxide levels. Additionally, knockout mice exhibited heightened oxidative and nitrosative stress, along with impaired vascular function, especially under diabetic conditions. FXYD1 protects endothelial NO synthase from redox dysregulation, offering protection against both hypertension and diabetic vascular oxidative stress [109]. In this regard, Cai et al. (2023), using molecular biology tools, genetic engineering, and metabolic analyses to investigate the role of the Na, K‒ATPase (NKA)-Src axis in mitochondrial function in human-induced pluripotent stem cell-derived cardiomyocytes (hiPSC-CMs), demonstrated that Na, K‒ATPase α1/Src regulation plays a role in the tonic stimulation of mitochondrial metabolism and ROS production. This has potential implications for addressing mitochondrial dysfunction in cardiometabolic diseases [110].

Most FXYD proteins protect the β1 subunit against glutathionylation, an oxidative modification that destabilizes the heterodimer and inhibits Na, K‒ATPase activity. The point mutation in the cysteine residue of FXYD3 increases sensitivity to oxidative stress induced by the chemotherapeutic doxorubicin and γ-irradiation [111,112].”

Comment 10: Please represent the chemical structures in Figure 3 with simple skeletal structural formulas.

Answer 10: The figure 3 of the original manuscript was redesigned to cope with referee suggestion (page 8 of revised manuscript).

Reviewer 3 Report

Comments and Suggestions for Authors

1. I suggest that the authors include more recent studies published within the last 3–5 years. While the manuscript references key foundational works, adding newer findings, particularly on Na,K-ATPase in specific cancer types like pancreatic or lung cancer, would enhance its relevance and timeliness.

2. I think the authors should clearly define the scope of the review in the introduction. For example, specifying whether the focus is on the therapeutic potential, molecular mechanisms, or both.

3. The authors need to address discrepancies in the literature regarding the role of Na,K-ATPase in cancer progression versus suppression. 

4. I suggest including a section on emerging therapeutic strategies targeting Na,K-ATPase, such as the development of isoform-specific inhibitors or nanotechnology-based drug delivery systems. 

5. I think the authors should enhance the manuscript with schematic diagrams summarizing key pathways involving Na,K-ATPase and its role in cancer. 

6. The authors need to provide a comparative table summarizing the role of Na,K-ATPase across different cancer types. 

7. I suggest discussing methodological limitations in studying Na,K-ATPase in cancer.

8. The authors need to consider incorporating insights from disciplines such as bioinformatics or structural biology. 

9. I think the authors should expand the discussion on future research directions.

Author Response

Comment 1: I suggest that the authors include more recent studies published within the last 3–5 years. While the manuscript references key foundational works, adding newer findings, particularly on Na,K-ATPase in specific cancer types like pancreatic or lung cancer, would enhance its relevance and timeliness.

Answer 1:  We understood the referee´s concern regarding pancreatic and lung malignant tumors and some new adds on the manuscript text were performed. All insertions were based on recently published articles as specifically required and were as follows:

            “FXYD3 (antigen-8 of mammary tumor) is known to exert protection of the β1 subunit against glutathionylation, an oxidative modification that destabilizes the α-β heterodimer and inhibits Na, K‒ATPase activity. A specific cysteine (Cys) residue of FXYD proteins seems to be crucial for such protection. One of the FXYD proteins, FXYD3, confers chemotherapy resistance when it is overexpressed in cancer cells. Chemotherapeutic compounds as doxorubicin can induce oxidative stress, and Liu et al. 2022 shown that suppression of FXYD3 with siRNA in pancreatic and breast cancer cells which strongly express FXYD3, increased doxorubicin-induced cytotoxicity. This process restored the sensitivity of these tumor cells to one of the most used chemotherapics [112].

Bioinformatic analysis tools were determinant to evaluate the role of a high expression of KDM3A (lysine demethylase 3A) and DCLK1 (doublecortin-like kinase 1) and reduced expression FXYD3 in lung cancer. In the case of DCLK1, this article demonstrated its role in the FXYD3 suppression [280]. A reduced expression of FXYD3 was reported in lung cancer cells, in which its inactivation was identified as a key player in lung cancer progression [281]. In lung tumors, the increasing levels of DCLK1, promotes the proliferation and metastasis of lung cancer cells through the downregulation of FXYD3 [280].”

Comment 2: I think the authors should clearly define the scope of the review in the introduction. For example, specifying whether the focus is on the therapeutic potential, molecular mechanisms, or both.

Answer 2: We agree that our introduction section was a little short, but our primary intention was to assign that the main scope of our review was to focus on structurally oriented changes on the Na-pump driven by hormonal signaling, protein kinases and regulatory protein-protein interactions (FXYD regulation). The therapeutic potential emerged as a colateral information throughout the manuscript text development and for this reason we performed no changes in this regard on the text of intro. However, we agree that the concept of molecular mechanisms permeates the entire manuscript. Thus, we add one statement about this, rewriting the end of this section as follows:

“The aims of this review are to correlate the myriad of functional changes in the Na, K‒ATPase structure upon hormonal regulation and the actions of protein kinases and FXYD proteins to identify the potential implications of these enzyme modifications in health and pathophysiology. It is primarly important to us to describe in good detail the molecular mechanisms underlying the events which leads to the regulation of the Na-pump to shift from a homeostatic physiological status to become a player in the origin and development of several diseases.”

 Comment 3: The authors need to address discrepancies in the literature regarding the role of Na,K-ATPase in cancer progression versus suppression.

Answer 3: We think that this query is answered throughout the revised text of the cancer section and also in the content of the new table which was inserted to answer the question number 6 of the referee.

Comment 4:  I suggest including a section on emerging therapeutic strategies targeting Na,K-ATPase, such as the development of isoform-specific inhibitors or nanotechnology-based drug delivery systems.

Answer 4: We agree with this referee suggestion and we included some paragraphs in the revised manuscript text. This addendum describes some promising therapeutic strategies based on nanoparticles, which are employed as efficient tools to improve drug delivery systems on cancer therapies. This amendment goes as follows:

“Inhibition of Na, K‒ATPase through an isoform-selective fashion has been described as a promising strategy for cancer treatment due to its critical role in maintaining cellular functions. Advancements in nanotechnology permit the creation of targeted drug delivery systems that enhance the efficacy and specificity of Na, K‒ATPase inhibitors, minimizing side effects and improving therapeutic profiles. DSPE-PEG nanocarrier particles were conjugated with a peptide targeting Na, K‒ATPase α1 with the task to deliver Epirubicin (EPI) specifically to breast cancer cells, reducing the size and volume of breast tumors. This targeted approach allows for the slow release of EPI within cancer cells, significantly inhibiting proliferation and migration concurrently reducing systemic colateral effects [253]. In addition, Doxorubicin (DOX)-encapsulated nanoparticles with poly(lactic-co-glycolic acid) (PLGA) in composition and fusioned with a 13-amino acid peptide targeting Na, K‒ATPase exhibited enhanced cellular uptake and antitumor activity in breast cancer models. This strategy improved the main survival rate of tumor-bearing mice and decrease systemic toxicity [254].

Ultra-small vanadate prodrug nanoparticles have been developed to selectively inhibit Na, K‒ATPase in cancer cells. This strategy is also known as stimuli-responsive nanoparticles. This kind of nanoparticle is modified with tannic acid and bovine albumin with the goal to reduce systemic toxicity. This nanoparticle is also sensitive to reactive oxygen species generated at the tumor sites. Near-infrared (NIR) photothermal properties further enhance the inhibition of Na, K‒ATPase, causing a considerable cancer cell death with a very low impact on healthy tissues [255].  Furthermore, other therapeutic strategies involving vanadate complexes have been described as antitumoral agents. Vanadate derivatives influences lipid peroxidation, induces changes in the cell cycle, and promotes the generation of ROS. This inhibitor also affects several signaling pathways, such as AMPK and phosphatase 1B, in breast cancer [256,257].

Some pH-responsive drug delivery nanocarriers were designed to take advantage from the acidic microenvironment of tumors, starting to release drugs only at the tumor site. This strategy increases drug uptake by cancer cells, reducing off-target effects, permitting the resolution of some challenges such as poor tumor selectivity and multidrug resistance [258]. In this context, we can remark that nanomedicine strategies aim to overcome intrinsic and acquired drug resistance by increasing intracellular drug accumulation and selectivity. Multifunctional nanoparticles can also deliver drug combinations employed on synergistic treatments, ameliorating the quality of therapeutic regimens overcoming limitations of conventional chemotherapy [259–261].”

Comment 5: I think the authors should enhance the manuscript with schematic diagrams summarizing key pathways involving Na,K-ATPase and its role in cancer.

Answer 5: Many thanks for your comment, we introduced two originally designed figures to illustrate some key events and the cellular signaling linked to involvement of Na, K‒ATPase on cancer (figure 5, page 31, and figure 6, page 36).

Comment 6: The authors need to provide a comparative table summarizing the role of Na,K-ATPase across different cancer types.

Answer 6: We appreciate this suggestion and we generate one new table encompassing the correlation with enzyme and FXYD isoforms with several types of cancer, pointing out their expression and tumoral prognosis (Table 1 of revised text).

Comment 7: I suggest discussing methodological limitations in studying Na,K-ATPase in cancer.

Answer 7 To address this comment we introduce several paragraphs at the page 32 of the revised cancer section of the manuscript, properly discussing this point, and it goes as follows:

“Recent research on Na, K‒ATPase in cancer has revealed its pivotal role in several cellular processes, including cell adhesion, migration, invasion and intracellular signaling [247,248]. However, methodological limitations pose challenges in fully understanding its role and influence in cancer setting and progression. Thus, the study of Na, K‒ATPase in cancer is hindered by complex regulatory mechanisms of signaling, differential expression of α, β and γ isoforms, and the interplay with a lot of cellular processes as inflammation, production of cytokines and the influence of the activation of oncogenes.

Analysis of Na, K‒ATPase α- and β-subunit expression in bladder cancer samples shown that decreased enzyme expression levels are linked with the increase of recurrence risk. This suggests that Na, K‒ATPase subunit expression might serve as a potential predictor of clinical outcomes [249,250]. Moreover, liver and colorectal cancer metastases exhibit differential expression of specific isoforms of Na-pump (α1, α3, and β1), with implications for their role in the control of ion gradients and support of nuclear enzyme functions involved in mitosis control [251]. In this context, the α3 isoform was predominantly located in the cytoplasmic membrane in healthy colon and lung cells. However, the distribution of this isoform was shifted to a predominantly peri-nuclear location in several tumors [252]. It was shown that in liver metastasis the α3 isoform was mainly detected at a peri-nuclear location and was more diffusely expressed across the cytoplasm of tumor metastatic cells. Thus, it is evident that the differential expression and function of Na, K‒ATPase isoforms in cancer cells versus normal cells complicate the full comprehension of its role in cancer [248,251].

Future research needs to elucidate the specific contributions of each Na, K‒ATPase isoform. In summary, the α3β1 isozyme may have potential as a novel exploratory biomarker for metastases cells. However, further studies still need to be done to confirm and expand this apparent involvement.

Voltage-gated sodium channels (VGSCs) and Na, K‒ATPase are co-regulated by inflammatory mediators in metastatic breast cancer. This co-regulation is sodium-dependent, and VGSCs are required for the inflammatory mediated effect on Na, K‒ATPase [247]. The study of inflammatory mediators like TNFα and Prostaglandin E2 on Na, K‒ATPase expression in cancer cells shows variability in RNA expression across different cell lines, indicating complex regulatory mechanisms that are not yet fully understood.

The α1 subunit of Na, K‒ATPase is intimately associated and regulates the proto-oncogene Src kinase, increasing aerobic glycolysis and tumor growth. Loss of this regulation is associated with increased lactate production and progressive tumor growth, highlighting the Na, K‒ATPase/Src kinase complex as a key player as a tumor suppressor [253].

In spite of these several challenges, understanding Na, K‒ATPase roles in cancer progression and treatment of resistance has potential for development of therapeutic strategies. Further research is crucial to overcome these methodological limitations and fully exploit the Na, K‒ATPase as a target in cancer therapy.”

 Comment 8: The authors need to consider incorporating insights from disciplines such as bioinformatics or structural biology.

Answer 8: A deeper read of the full text of manuscript will evidence that during several sections the text the context of discussion permeates information which was directly obtained from biosciences as Bioinformatics and also from Structural Biology approaches as crystallization or NMR. Some examples of our point-of-view are listed below:

  • Page 5, figure 1 We describe and discuss high-resolution crystals of the pump enzyme;
  • References 57 and 245 discussed on several point of the manuscript, used the SAGE tool from NIH (a comparative bioinformatic resource which was useful to gather information regarding differential expression of proteins during evolution of healthy to tumoral cells);
  • Newly added references 280 and 281 directly discuss the employment of bioiformatic approaches to discover the role of FXYD3 and DLCK1 during the evolution of lung cancer.

 Comment 9: I think the authors should expand the discussion on future research directions.

Answer 9: We agree with the referee suggestion and we introduced a couple of paragraphs at the end of cancer section of the manuscript to expand the discussion with some potential future directions regarding this section of the review. This addendum goes as follows:

“Future directions on Na, K‒ATPase and cancer research

Some challenges still surround the future of Na, K‒ATPase related cancer research, in which the optimization and precise targeting associated with better control of drug release might be viewed as the main promise of nanomedicine. If this directions would be fully achieved it would be increase the specificity and toxicity of existing nanoparticle-based systems, improving therapeutic efficacy and clinical outcomes [261]. In spite of current intrinsic potential of nanocarriers, only a few real nanomedicine drugs have received FDA approval to being employed at cancer treatments. Continued research and clinical trials are necessary to validate the safety and effectiveness of novel nanocarriers and to facilitate their translation into clinical practice [260].

Na, K‒ATPase is very well known as a target for cardiotonic steroids, which show promise in overcoming multi-drug resistance in cancer cells. However, developing of resistance to these compounds and their precise mechanisms of action require further investigation [282]. Combined treatments employing the synergistic effect of CTS and chemotherapies, in association with multidrug deliverance systems would be very useful in future cancer research [231]. Further research and clinical validation are essential to realize the full potential of these innovative therapeutic strategies.”

Round 2

Reviewer 2 Report

Comments and Suggestions for Authors

After careful reconsideration, the corrected manuscript met my expectations.

I think now, that the manuscript is suitable for publication.  

Reviewer 3 Report

Comments and Suggestions for Authors

No more comments